# Programmable spatial deformation by controllable off-center freestanding 4D printing of continuous fiber reinforced liquid crystal elastomer composites

Qingrui Wang[1], Xiaoyong Tian [1] ✉, Daokang Zhang[1], Yanli Zhou[1], Wanquan Yan[1] & Dichen Li[1]

Owing to their high deformation ability, 4D printed structures have various applications in origami structures, soft robotics and deployable mechanisms. As a material with programmable molecular chain orientation, liquid crystal elastomer is expected to produce the freestanding, bearable and deformable three-dimensional structure. However, majority of the existing 4D printing methods for liquid crystal elastomers can only fabricate planar structures, which limits their deformation designability and bearing capacity. Here we propose a direct ink writing based 4D printing method for freestanding continuous fiber reinforced composites. Continuous fibers can support freestanding structures during the printing process and improve the mechanical property and deformation ability of 4D printed structures. In this paper, the integration of 4D printed structures with fully impregnated composite interfaces, programmable deformation ability and high bearing capacity are realized by adjusting the off-center distribution of the fibers, and the printed liquid crystal composite can carry a load of up to 2805 times its own weight and achieve a bending deformation curvature of 0.33 mm$^{-1}$ at 150 °C. This research is expected to open new avenues for creating soft robotics, mechanical metamaterials and artificial muscles.

4D printing is a type of additive manufacturing technology used in the fabrication of smart structures that can produce controllable deformation in response to external stimuli. Shape memory materials[1], anisotropic materials[2] and composites[3] have been widely used as raw materials for 4D printing. According to their deformation principle, shape memory materials can rapidly change from a temporary shape to their final shape at the critical temperature[4,5]. However, the disadvantage of these materials is that the deformation process is discontinuous[6] and difficult to replicate[7]. Moreover, anisotropic materials such as liquid crystal elastomers (LCEs) are also used for 4D printing. LCEs are smart materials with high deformation ability[8], relatively significant mechanical performance[9–11] and reversible

deformation process that can be prepared by direct ink writing (DIW) 4D printing[12,13], and can respond to various external stimuli, such as light[14], humidity[13,15], and heat[16]. LCEs are composed of orientated liquid crystalline polymers that have high thermal shrinkage effect along the direction of the molecular chain[17], and the orientations of the LCE molecular chains determine the mechanical property of the polymers[18,19]. In addition, composites are also used for 4D printing. The design method for 4D printing composite structures is inspired by the growth of pine cones[20–22] and the movement of wheat awn[23] in nature, whose deformations depend on their anisotropic expanded cell walls caused by the existence of embedded oriented stiff cellulose fibrils. Similarly, 4D printed composite structures are typically composed of

[1]State Key Laboratory for Manufacturing Systems Engineering, Xi'an Jiaotong University, Xi'an 710049 Shaanxi, China. ✉e-mail: leoxyt@mail.xjtu.edu.cn

two materials with different expansion coefficients distributed in gradient and can produce bending deformation under external stimulation. The deformation amplitude and direction of the composite structure can be controlled by the programmable orientation of reinforcing components, such as embedded particles[24], continuous fibers[25], and oriented polymer chains[26]. Due to the diversity of materials, 4D printing structure has various deformation stimulation conditions. Specifically, external stimuli that can induce deformation include temperature[27], light[28,29], humidity[30], and electromagnetic field[31]. Due to their deformation performances, 4D printed structures can be widely used in various fields, such as aerospace[32], soft robotics[33,34] and artificial muscles[35].

As a material with programmable molecular chain orientation, LCE is expected to produce the freestanding, bearable and deformable three-dimensional structure. However, 4D printing with LCEs and other active materials still faces some challenges with regard to the deformation characteristics. As widely used 4D printing materials, anisotropic materials and composite structures can produce continuous deformation, but their deformation mode is relatively simple[36,37]. For example, 4D printed structures with deformable composites are typically thin bilayer structures because their deformation mode involves bending deformation driven by two materials with different coefficients of thermal expansion (CTEs). Hoa et al.[38–40] prepared laminates composed of materials with different CTEs by 4D printing to realize the moldless composites manufacturing. In addition, anisotropic LCE structures are usually 1D[41] or 2D structures[42] because the deformation of polymers depends on the orientation degree of anisotropy and the schemes that can control the orientation of molecular chains or reinforced particles are mainly to control the movement path of the nozzle along the $x$ and $y$ directions[12]. For these reasons, most 4D printing methods for LCE structures or bilayer composites are either planar structures[43] or simple shapes obtained by folding[44], curling[37] and accumulation[45] based on planar structures, which significantly limits the shape design of 4D printed structures. Several efforts have been made to obtain more complex deformable structures by 4D printing. Guo et al.[46] prepared three-dimensional LCE structures whose molecular orientation can be controlled locally, however, the voxelated structure limits the 4D printing efficiency.

Another critical challenge for 4D printed structures is their poor mechanical bearing capacity caused by the material and shape restrictions. However, efforts to improve the mechanical property of 4D printed structures typically weaken or sacrifice their deformation capacity. Liu et al.[27] reported that the mechanical properties of 4D printed precursor materials were improved after heat treatment, however, the obtained ceramic materials could not be deformed again. According to Zeng et al.[47], embedding chopped or continuous fibers into polymer matrices can improve the mechanical properties while retaining the appropriate deformation capacity. Moreover, the shapes of 4D printed structures with composites or anisotropic materials is typically another factor limiting their mechanical bearing capacity. Most of the existing 4D printed LCE structures are linear or planar and can only produce tensile stress during shrinkage deformation[42], which prevents them from maintaining their structural stability under the external compression forces[48–50]. Although shape memory materials can be used to prepare complex and deformable three-dimensional structures with an actuating ability, they can only produce a deformation response within a very small range near the critical temperature, and the deformation process is usually irreversible[36]. Therefore, the application scenarios for 4D printing are limited.

To solve these problems, a 4D printing method for freestanding LCE composite truss structures is required. The high bearing capacity of the three-dimensional structures comes from the mechanical reinforcement in three directions, and the deformation controllability is based on the bendable trusses. Peng et al.[51]. realized the 4D printing of the freestanding LCE structures by using multiple laser sources to cure the LCE in-situ when the nozzle moves and extrudes the liquid crystal material. However, this method has the following limitations: (i) the forming process of LCE structures depends on supporting structures; (ii) the LCE truss structures have little bearing capacity; (iii) the solidification of liquid crystal materials requires expensive high-power multiple laser sources. The composite truss with embedded and controllable off-center continuous fibers is expected to achieve the integration of the mechanical bearing capacity and deformation function simultaneously.

In this study, a continuous fiber direct ink writing (CFDIW) 4D printing method based on off-center continuous fiber reinforced LCEs (CFRLCEs) has been proposed. The introduction of continuous fibers can make the structure preparation process more convenient and improve the deformation capacity[52] and mechanical properties[53] of the LCE structure. Moreover, the mechanical properties of 4D printing structures can be improved because the elastic modulus of continuous fibers is much higher than that of LCEs and other deformable polymers. In terms of the deformation performance, the difference in the CTEs of the continuous fibers and LCE matrix induces the change in the deformation mode of the pure liquid crystal from shrinkage deformation to bending deformation, which further increases the deformation capacity. Depending on the support ability of the continuous fibers, composites can be directly prepared into three-dimensional structures instead of preparing planar structures via general 4D printing, and these three-dimensional structures have improved mechanical properties. In addition, the CFDIW process allows the adjustment of the off-center position of the fiber in the composite filament, which allows for the easy control of the bending deformation direction of the composites, a better impregnation effect between fibers and liquid crystals, and production of considerable deformation in prepared three-dimensional structures.

## Results

### CFDIW 4D printing method for off-center CFRLCEs

During CFDIW 4D printing process, the formula comprising liquid crystal monomers, crosslinkers, and photoinitiators was used as the ink for preparing the LCE matrix, and their molecular structures are shown in Fig. 1b. For imaging purposes, fluorescent agents were added to the composition. The synthesis mechanism of LCE is shown in Supplementary Fig. 1, illustrating that the liquid crystal monomers form polymer chains under the heating condition and crosslinked under ultraviolet light to form cured LCEs[54]. The formula materials were subjected to differential scanning calorimetry (DSC) to determine their addition reaction temperature (Supplementary Fig. 2), and Fourier transform infrared spectrometer (FTIR) tests showed that the polymerization was almost fully carried out after heating the samples at 160 °C for 30 min (Supplementary Fig. 3). In addition, the DIW process requires the liquid crystal material to be shear-thinning so that it can be smoothly extruded from the nozzle. The shear viscosity of liquid crystal components were tested at different temperatures, and the results, which indicate the occurrence of an ideal shear thinning effect, are shown in Fig. 1a. From the figure, a lower printing temperature leads to higher shear viscosity of the materials, making the liquid crystals easier to adhere to the fiber surfaces. The continuous fibers used in this paper were aramid fibers due to their shear resistance. Since the formation of the 4D printed structures depends on the support of the fibers and does not require the liquid crystals to attach to the platform, the liquid crystal do not need to have a high shear elastic modulus such as the current DIW processes[55]. See *Methods* for details of raw materials.

In order to prepare CFRLCEs, the continuous fiber bundle needs to be fully impregnated by liquid crystal polymers and adjusted to the off-center position of the composite filament, and then the composite is cured under ultraviolet light. Figure 1c shows the CFDIW process in which the composite goes through three stages of impregnation,

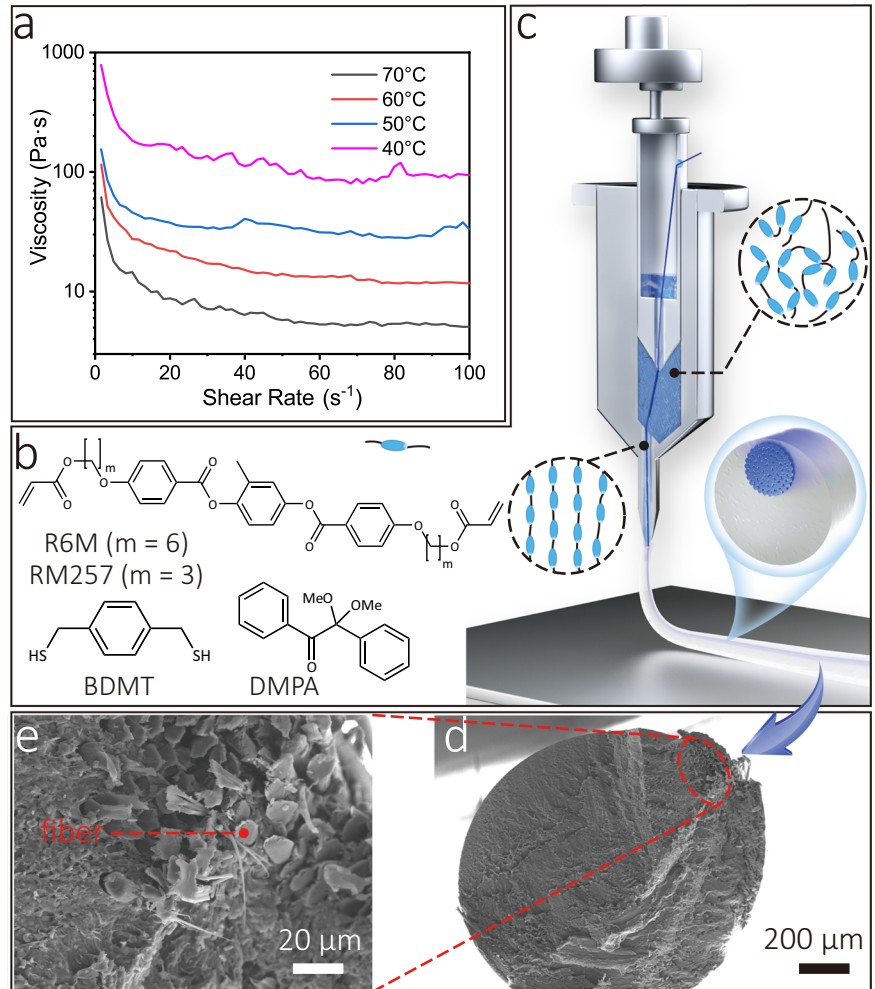

**Fig. 1 | Raw materials, mechanism, and extruded composite LCE filaments of the CFDIW Process. a** Shear viscosity of the liquid crystal material components. **b** Chemical structures of liquid crystal monomers, crosslinkers, and photoinitiators. **c** The mechanism of the CFDIW process, including three stages of impregnation, extrusion and curing. **d** The low magnification SEM image of the cross section of the composite, indicating the off-center distribution effect of the fiber bundle. **e** The high magnification SEM image of the cross section of the composite, indicating a sufficient impregnation between the two materials.

extrusion and curing. The fibers contact with liquid crystals in the chamber, and the liquid crystals are extruded out of the chamber while the fibers are pulled out of the nozzle to obtain a uniform CFRLCE composite filament. Since both ends of the fiber bundle are tensioned, the fibers are subjected to a literal force provided by the nozzle and thus leads to an off-center distribution of fibers. The cross section of the filament observed by scanning electron microscope (SEM) is shown in Fig. 1d, which proves the existence of the off-center distribution effect. In the process of composite forming, the off-center distribution is helpful for liquid crystals to impregnate on the fiber surface. The higher off-center effect makes the cross-sectional shape of the fibers more flat, so that the liquid crystals are easier to penetrate into the interior of the fiber bundle. The cross section of the pulled composite with a high off-center effect was observed by SEM with a higher magnification, as shown in Fig. 1e, indicating that the impregnation of the two materials was sufficient. In contrast, when the lateral force applied by the nozzle is low during the pulling process, the cross section of the fiber is close to a circle, and there are some cavities not filled with liquid crystal inside (Supplementary Fig. 4). The interfacial properties were characterized by utilizing the micro-droplet debonding test, and the CFRLCE composites exhibit a maximum increased interfacial shear strength (IFSS) reaching up to 5.58 MPa. In addition, it is necessary to prevents liquid crystals from accumulating droplets on the fiber bundle surface to obtain the filament with a uniform shape

during the curing process. The average wetting angle between the liquid crystals and continuous aramid fibers is 32.1° at room temperature, as shown in Supplementary Fig. 5, indicating that liquid crystals have the ability to spread on the fiber bundle surface. In order to achieve good thermal deformation effect of 4D printed composite filaments, the molecular chains of the extruded liquid crystals require uniform orientation. The LCEs of the pulled composite filaments were tested by 2D X-ray diffraction (XRD), as shown in Supplementary Fig. 6, illustrating that the liquid crystal polymer chains are oriented. Owing to the huge difference of CTEs between fibers and liquid crystals, composites will produce bending deformation when heated. A simulation analysis was performed using ABAQUS, revealing that the off-center composite filament structure had a good thermal bending deformation effect at 130 °C, as shown in Fig. 2c. The process of rapid deformation of the off-center composite filament placed in a container with an ambient temperature of 150 °C was recorded (Supplementary Movie 1), and the frame images at $t = 0$ s, 4 s and 23 s in the video are shown in Fig. 2d–f, illustrating a high thermal-induced bending deformation curvature of the composite.

Compared with existing DIW 4D printing equipment, CFDIW equipment has some structural differences in the piston and chamber. For most DIW equipment, there is no gap between the chamber and piston to allow the extrusion of the liquid crystal from the nozzle when the piston is pushed. However, the continuous fibers are required to

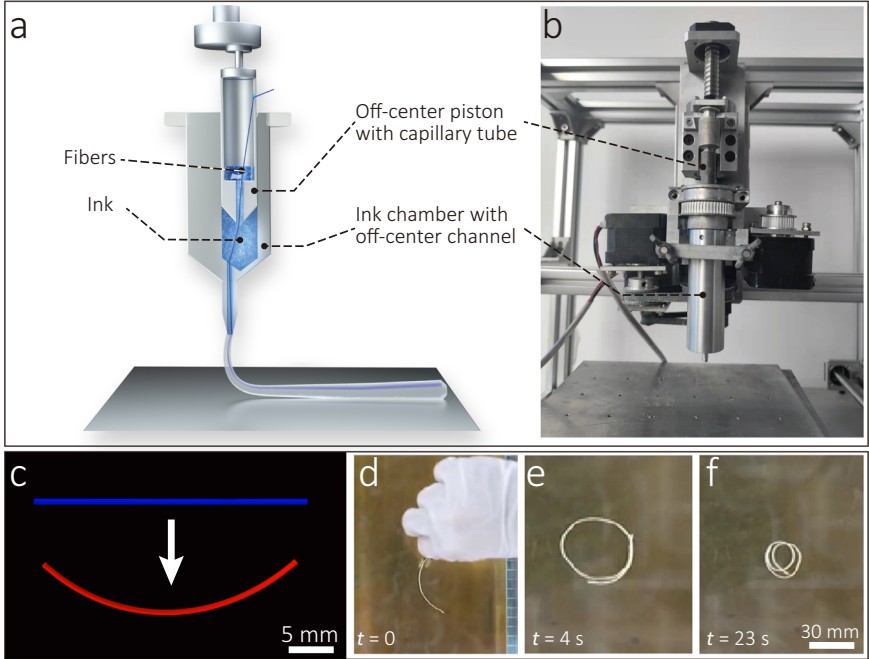

**Fig. 2 | Preparation of off-center CFRLCE materials. a** Schematic of the CFDIW equipment. **b** The Piston and the ink chamber of the CFDIW equipment. **c** Simulation analysis of the thermal deformation effect of the LCE composite. **d**–**f** The deformation states of a CFRLCE filament at an ambient temperature of 150 °C for t = 0 s, 4 s, and 23 s.

pass through the ink chamber and become fully impregnated with the liquid crystal ink while printing the continuous fiber reinforced LCE composites. Therefore, a semi-seal ink chamber with a capillary structure at the top was designed, and the liquid crystal ink was contained in both the chamber and capillary structure, which can prevent air from entering the chamber, and ensure the free movement and full impregnation of the continuous fibers, as shown in Fig. 2a and b. In addition, to make the fiber/LCE composite structure produce bending deformation under temperature stimulation, the distribution of the continuous fibers in the composite should be off-center. A coupling that can rotate relative to the axis of the nozzle was added to the 4D printing device to control the rotation of the ink chamber and adjust the off-center position of the fiber, as shown in Fig. 2a and b. The composite can be bent in any direction in the three-dimensional space by adjusting the off-center position of the fiber bundle on the composite filament section.

### Controlling the off-center degree of the fibers

In CFDIW 4D printing, the off-center degree of the fiber bundle is adjustable and affects the deformation curvature of the composite, and the most effective control scheme involves adjusting the inclination angle θ, as shown in Fig. 3a, and b, of the nozzle motion path. The nozzle moves like a spider and "spits out" the continuous fibers attached with liquid crystal during printing process. One end of the "spits out" fiber has been solidified in the previous printing step, and its another end is tensioned by the 4D printer. Therefore, The inclination angle, which can be positive, negative, or zero, is used to determine the lateral force on the fiber during the fiber-pulling process. Subsequently, the cross-sectional shape of the fiber bundle, which was later proved to be an important factor affecting the thermal deformation ability of composites, is affected by the lateral force on the fiber bundle. To observe the cross-sectional shape of the composite filaments, fibers were pulled out of the nozzle together with the liquid crystal at three inclination angles of −10°, 30° and 90°, and the cross sections of the cured filaments were imaged using SEM, as shown in Fig. 3f–h. When the tensile inclination angle was 90°, the fiber was

hardly subjected to any external lateral force exerted by the nozzle, so the cross-sectional shape of the fiber bundle was approximately a circle. With the decrease in the inclination angle to a negative value, the external force on the fiber in the lateral direction gradually increased, and the section shape of the fiber bundle became higher in aspect ratio.

In order to analyze the influence of cross-sectional shape of the fiber bundle on the thermal deformation of CFRLCE filaments, the curvature change of core-shell composite structure with off-center fiber distribution was calculated. When the lateral force is close to zero, the fiber section shape is approximately circular, and the deformation curvature of the composite material is as follows (detailed in Supplementary Discussion):

$$k = \frac{4E_R E_r R^2 r^2 (R-r)(\alpha_r - \alpha_R)\Delta T}{(E_R R^4 + E_r r^4)(E_R R^2 + E_r r^2) + 4E_R E_r R^2 r^2 (R-r)^2} \tag{1}$$

where $E_R$ and $E_r$ represent the moduli of the LCE and fiber respectively, $\alpha_R$ and $\alpha_r$ represent the CTEs of the LCE and fiber respectively, $T$ represents the ambient temperature, and $k$ represents the deformation curvature of the composites. The bending curvature $k$ was used to characterize the deformation capacity of the composite filament. The influence of the fiber section shape on the bending deformation curvature of the composites was modeled and calculated. The relationship between the deformation curvature and aspect ratio of the fiber section shape is shown in Supplementary Fig. 7, which illustrates that the cross-sectional shape with a higher elliptical eccentricity corresponds to a higher thermal-induced bending deformation ability. In addition, with the increase of off-center degree of the fibers, the cross section of the fibers with larger aspect ratio makes the cavities inside the fiber bundle disappear and leads to better impregnation effect. This will bring better deformation ability, because the bending deformation comes from the stress between the fiber and the liquid crystal with different thermal-induced shrinkage or expansion strains, and a better interface bonding characteristic is obviously conducive to the transfer of stress between the two materials.

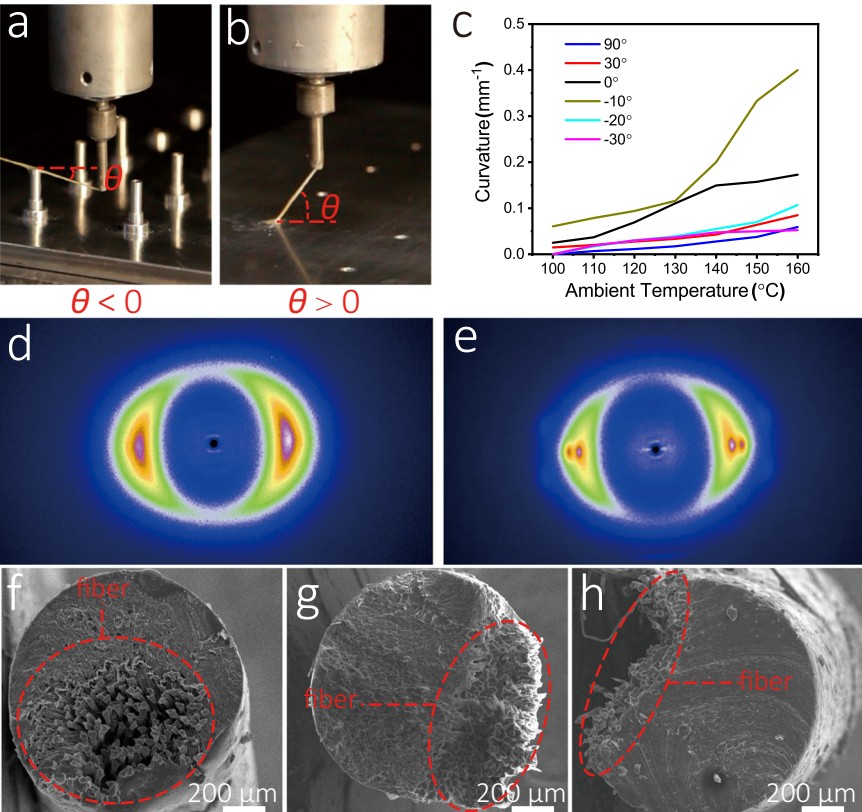

**Fig. 3 | Controlling the deformation characteristics of LCE using the tensile inclination angle. a** Schematic of positive inclination angle. **b** Schematic of negative inclination angle. **c** Effect of the tensile inclination angle on the deformation curvature of composites. **d** The 2D XRD pattern of the composite with −10° tensile inclination angle. **e** The 2D XRD pattern of the composite with 90° tensile inclination angle. **f** SEM image of composite with 90° tensile inclination angle. **g** SEM image of composite with 30° tensile inclination angle. **h** SEM image of composite with −10° tensile inclination angle.

To verify this theoretical analysis, the composite filaments with different tensile inclination angles were 4D printed and their deformation curvatures at different temperatures were tested, as shown in Fig. 3c. The deformation curvature of the composite with a tensile inclination angle of −10° was approximately one order of magnitude higher than that of the composite with a tensile inclination angle of 90°, indicating that the influence of the tensile inclination angle on the deformation effect was basically consistent with the theoretical results. However, when the tensile inclination angle continued to decrease, the deformation affect declined rather than improve, which may be because the excessively low tensile inclination angle destroys the forming quality of the 4D printed composite. Subsequently, whether the liquid crystal orientation was affected by the change in the tensile inclination angle was verified. The continuous fibers in the 4D printed filament composites with −10° and 90° tensile inclination angles were stripped off, and the remaining LCE parts were tested using 2D X-ray diffraction (XRD), as shown in Fig. 3d, and e (with tensile inclination angles of −10° and 90°). The liquid crystal polymer orientation of the two samples were found to be almost similar, where the orientation degrees are 0.20 and 0.19, respectively. Therefore, it can be inferred that the change in the tensile inclination angle has no significant effect on the molecular orientation, and the deformation curvature is almost completely determined by the shape of the fiber cross section, as shown in Fig. 3f–h.

### Controlling the off-center direction of the fibers

The advantage of the CFDIW 4D printing process is that it can control the formation structure of the truss structure and the distribution of the fibers in the truss simultaneously. If no additional control measures are adopted, the fibers will be distributed on the upper surface of the composite under the action of tension. However, the deformation of three-dimensional structure requires that the fibers are distributed at any position within the composite filament cross section, which can be achieved by adjusting the moving path of the nozzle. A composite truss with the fibers distributed on the left side was used as an example to demonstrate the control method of the off-center position. To form an adjustable off-center composite filament, the movement path of the nozzle includes pulling out, lifting and falling motions, as shown in Fig. 4a. The pulling out step was used to determine the fiber off-center direction while "spiting out" the composites, and the falling movement was used to determine the forming position of the truss. Specifically, the composite was pulled out of the nozzle with the fibers distributed at the top of the filament, and the filament was cured using an area light source except for its two ends. Then the nozzle was lifted along an arc trajectory, causing the filament to rotate in the x-z plane until it was vertically upwards, so that the fibers were distributed on the left side. Due to the two ends of the filament had not yet solidified, the rotational motion of the filament was not constrained. After the nozzle had fallen along an arc trajectory in the y-z plane, the end point of the fiber was finally cured using a higher-power point light source, and the off-center angle of the fibers had been adjusted (Fig. 4c). The theoretical analysis shows that the angle $\alpha$ between these two nozzle positions is approximately equal to the off-center angle $\beta$ of the fiber, as shown in Fig. 4b. Based on this relationship, a complete nozzle movement path was obtained. The 4D printing process of an L-shaped structure with adjustable off-center distribution positions of fibers is shown in Supplementary Movie 2. To verify the effectiveness of the off-center fiber control scheme, a filament structure comprising two sections with

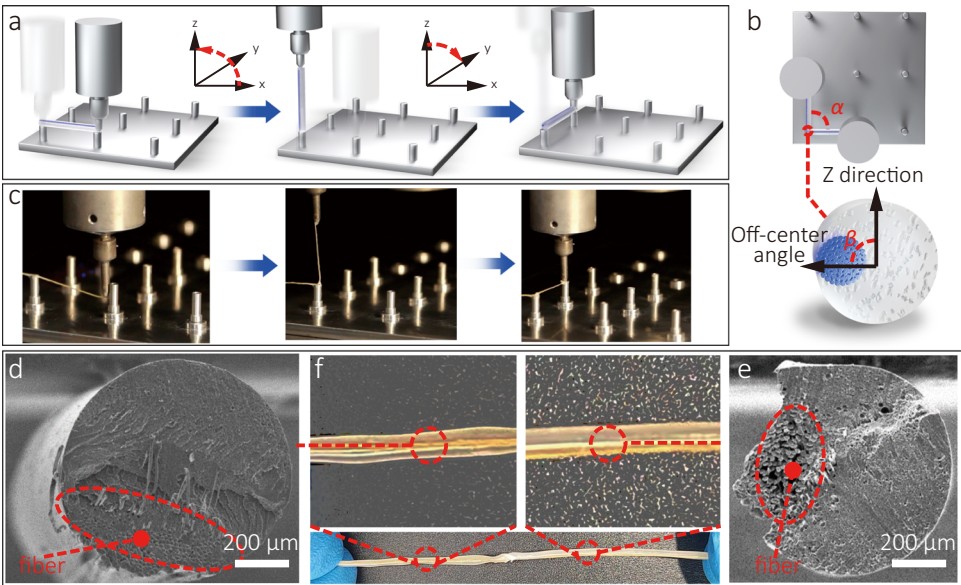

**Fig. 4 | Control method of fiber distribution position. a** Schematic of the 4D printing method for an off-center distributed composite filament. **b** 4D printing process of an off-center distributed composite filament. **c** Schematic diagram of the off-center angle. **d** SEM image of the cross section of a composite with fibers distributed on the top area. **e** SEM image of the cross section of a composite with fibers distributed on the right area. **f** A composite filament with various fiber off-center positions.

different fiber off-center positions was 4D printed (Fig. 4f) and an SEM test carried out on it, as shown in Fig. 4d and e. It can be observed that the fiber off-center angle can be accurately controlled as expected.

## Deformation properties of CFRLCEs

The effects of other 4D printing parameters on the deformation properties were also tested. The molecular orientations of the composite filaments with different ink extrusion speeds of 20 mm³ s⁻¹ and 80 mm³ s⁻¹ were tested by 2D XRD, as shown in Fig. 5a, where the orientation degree of the LCE extruded at a speed of 20 mm³ s⁻¹ was only 0.096. The molecular orientation has a significant influence on the thermal shrinkage property. Therefore, the higher the extrusion speed, the better is the deformation performance of the composite structure, as shown in Fig. 5a. Theoretically, the slight decrease in the impregnation effect caused by the increase of printing speed will weaken the deformation ability. However, compared with the improvement of the orientation of the liquid crystal molecular chain, the impregnation effect is obviously not a more important factor affecting the deformation of the composite. It can be expected that if the printing speed continues to be significantly increased, the deformation performance will seriously descend. In addition, the effect of printing speed on the curing time of liquid crystal can also be ignored.

The increase in the printing temperature reduces the deformation curvature of the composite truss (Fig. 5b) because the cooling time of high-temperature composite filaments is longer, so the LCE molecules are fully restored to the disordered orientation. The parameter experiments indicated that the optimum printing temperature was 30 °C, and that the extrusion of the liquid crystal from the nozzle is difficult at a lower printing temperature owing to its high viscosity. The light curing time is an important factor for determining the cross-linking degree of LCE (Fig. 5c), and subsequently affects the deformation ability and the mechanical property of LCE. The longer the light curing time, the higher is the crosslinking degree of LCE, which subsequently results in a higher mechanical modulus and mechanical strength. Figure 5d shows the mechanical properties of five groups of composite samples with light curing time of 10 min, 30 min, 1 h, 1 d and 5 d, respectively. However, a higher crosslinking degree limits the deformation ability of the composites. This confirms the feasibility of

the method in controlling the mechanical properties and deformation ability of composites. In comparison, the tensile modulus of LCE material without continuous fibers is only 0.24 GPa when the light curing time is 10 min, indicating the improvement of the mechanical property by continuous fibers. By optimizing the printing parameters outlined above, the LCE filament with printing parameters shown in Supplementary Table 1 achieved a bending deformation curvature of 0.33 mm⁻¹ at 150 °C.

## Controllable deformation ability of 4D printed trusses

The CFDIW printing process was more complex than that currently existing in additive manufacturing for printing truss structures, because it is necessary to simultaneously realize the controllability of the structural shape and internal fiber off-center position. This is evident in the M-shaped structure shown in Fig. 6a, which comprises four trusses in a plane. As illustrated in Fig. 4a, the formation of each truss requires the nozzle to pull out, lift, and subsequently fall, and the angle between the pulling out direction and the falling direction determines the fiber off-center position. The M-shaped truss structure was 4D printed based on the off-center position as shown in Fig. 6a and heated to 130 °C. Its shape changed from that in Fig. 6b to that in Fig. 6c, which is basically consistent with the ABAQUS simulation results shown in Fig. 6d. To verify the influence of the fiber off-center position on the deformation shape of the structure, two square structures with different fiber off-center positions were designed and 4D printed, as shown in Fig. 6e, f, i, and j. Note that because the fiber off-center angles of the truss in Fig. 6i were zero, the nozzle could directly fall on the target position without lifting. The deformation shapes and simulation results at the ambient temperature of 130 °C are shown in Fig. 6g, h, k, and l, indicating a different deformation result for the square structure. Therefore, even for 4D printing structures with the same shape parameters, the deformation processes of the structures can also be adjusted by the off-center positions of the continuous fibers, indicating a high deformation programmability of the CFDIW process.

Owing to the modulus of continuous fiber is much higher than that of LCE, the freestanding forming of cured trusses can be realized during the CFDIW 4D printing process. Two different laser light sources are needed to cure the LCE composites: A surface light source is

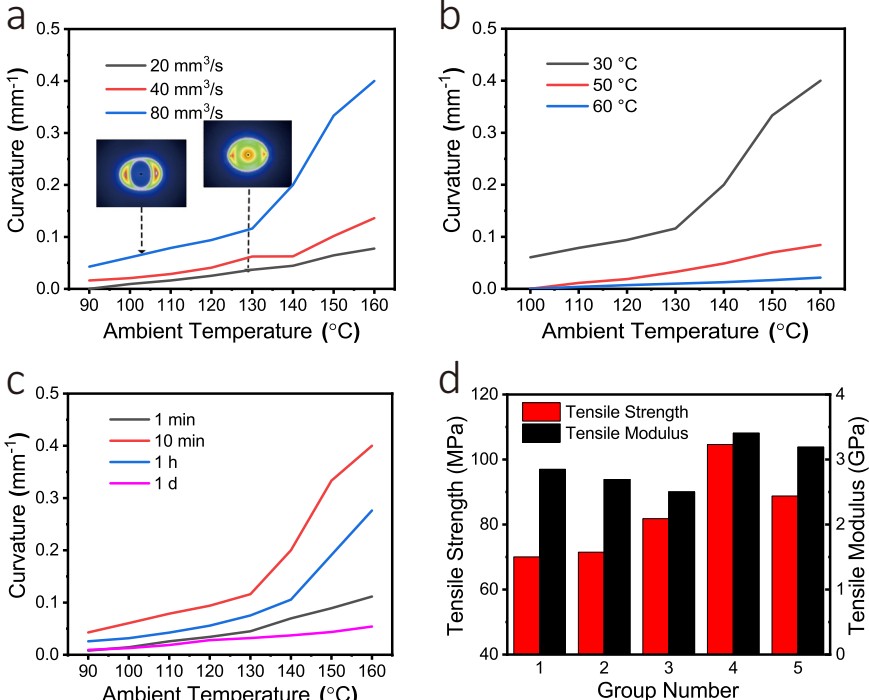

**Fig. 5 | Effect of 4D printing parameters on deformation performance of LCE.** **a** Effect of the extrusion rate on the deformation curvature of composite. **b** Effect of the printing temperature on the deformation curvature of composite. **c** Effect of the light curing time on the deformation curvature of composites. **d** Effect of the light curing time on the tensile performance of the composites.

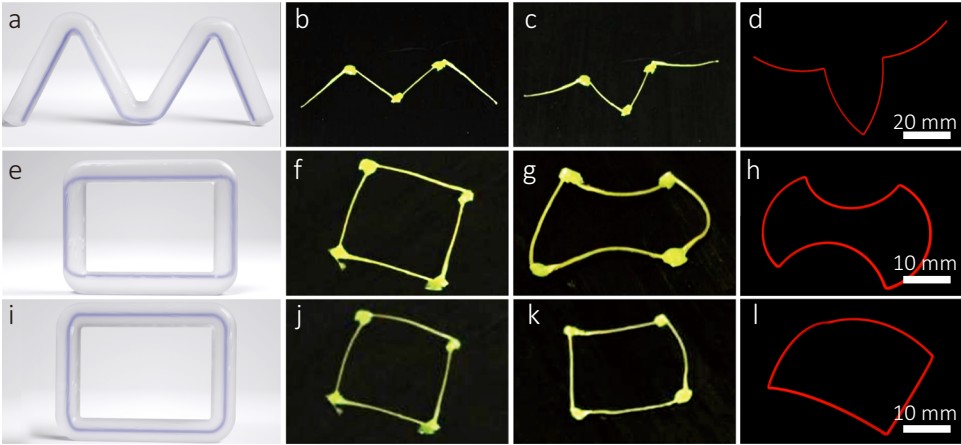

**Fig. 6 | Thermal-induced deformation of the CFRLCE trusses. a** Off-center position of the fibers in the M-shaped structure. **b** M-shaped structure before deformation. **c** M-shaped structure after deformation. **d** Deformation simulation results of the M-shaped structure. **e** Off-center position of the fibers in the square structure. **f** Square structure before deformation. **g** Square structure after deformation. **h** Deformation simulation result of the square structure. **i** Off-center position of the fibers in another square structure. **j** Another square structure before deformation. **k** Another square structure after deformation. **l** Deformation simulation result of another square structure.

used to cure the wire evenly, while a high-power point light source is used to cure the truss nodes. A triangular truss was used as an example to demonstrate the construction method for the truss structure. The piston of the printer was pushed to simultaneously extrude the liquid crystal and move the nozzle along the linear path with an inclination angle of $\theta$. Subsequently, a continuous fiber/LCE composite filament was formed because one end of the fiber was fixed onto the platform, and subsequent curing of the composite of Region 1 was done using an area light source (Fig. 7a–d). Because the end of the fiber in Region 1 supports the structure, it needs to be cured using a higher-power ultraviolet point light source. The nozzle was continuously moved

along an inclination angle of $\theta$ and a section of fiber pulled out before the fiber in Region 2 was cured. Here, ultraviolet light was prevented from irradiating the fiber midpoint between Region 1 and Region 2. Finally, the nozzle was moved along the arc path to ensure it falls until it stays on the platform. The remaining region was subsequently solidified using a point light source to complete the printing of the truss unit. The 4D printing process of the triangular truss is shown in Supplementary Movie 3. Furthermore, a freestanding pyramid shaped three-dimensional structure was printed without auxiliary support structures, as shown in Fig. 7f and g. The deformation result of the pyramid structure under heating is simulated by ABAQUS (Fig. 7e and

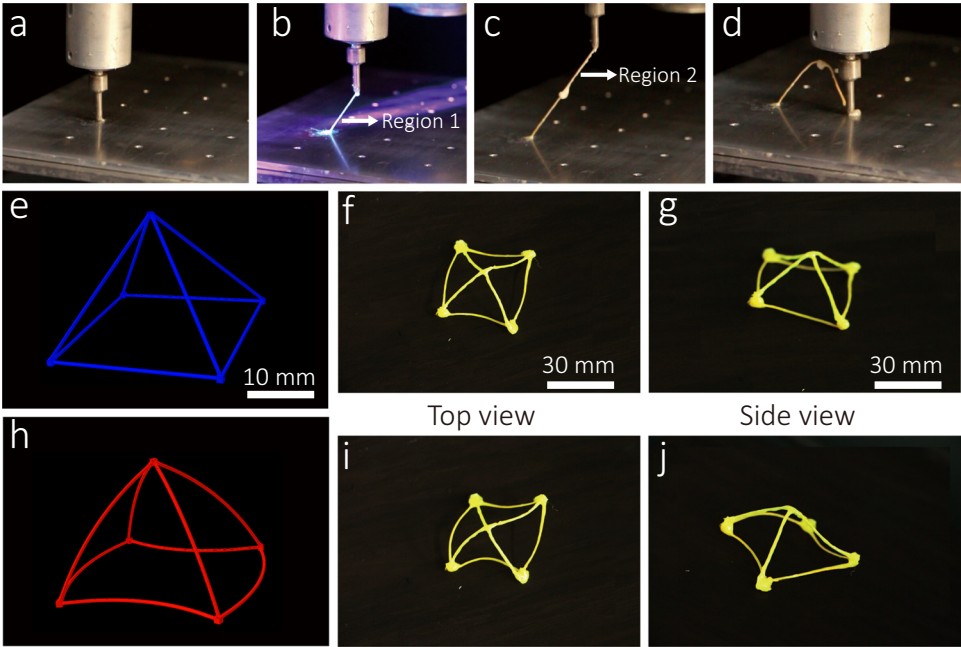

**Fig. 7 | 4D printed freestanding CFRLCE composites. a–d** 4D printing process for a triangular truss unit. **e** The schematic diagram of an undeformed pyramid structure. **f, g** Pyramid structure before deformation (top view and side view).
**h** Simulation of the thermal deformation state of the pyramid structure. **i, j** Pyramid structure after deformation (top view and side view).

h). When the structure was heated to 130 °C, the four trusses at the bottom shrunk due to the bending deformation, and the four trusses at the top twisted while shrinking. The final deformation result is shown in Fig. 7i and j.

### Bearing capacity of 4D printed trusses

The CFDIW process enables 4D printing for freestanding CFRLCE trusses, making it possible to increase the bearing capacity of composite structures. Owing to the CFDIW process is not easy to prepare composites with on-center distributed fibers, it is necessary to analyze the impact of fiber distribution on mechanical properties through simulation. When maintaining other printing parameters such as fiber content unchanged, the distribution of fibers hardly has a significant impact on the tensile property of composite filaments. Therefore, mechanical analysis of the bending behavior of the structure was emphasized. Here, pyramid structures are used for ABAQUS simulation analysis. The fibers inside the four filaments at the top of the pyramid structures are distributed on-center and off-center (with eccentric angles of 0 or 90°) respectively, and vertical downward compressing forces are applied to trusses at the top. When the applied displacement is one fifth of the height of the structures, the shapes of the structures are shown in Fig. 8a. It can be clearly seen that when fibers off-center distributed at the top, the truss produces almost no torsional behavior. When the fibers off-center distributed at the side or on-center, the truss undergoes severe torsion, which leads to earlier instability of the structures. The reason for the difference in mechanical behavior is that when the fiber off-center direction is the top of the filament, the larger aspect ratio of the fiber cross-section contributes to a better resistance to horizontal torque, and thus can effectively avoid structural failure due to torsion, while on-center or laterally off-center distributed fibers cannot resist this torsion. This theoretically proves that the off-center distribution of fibers can improve mechanical properties of the CFRLCE structures.

According to the results in Fig. 5d, the modulus of the LCE composite can be improved by increasing the light curing time. Therefore, a truss structure of four pyramid-shaped units was prepared as shown in Fig. 8c with a light curing time of 48 h, demonstrating a good

bearing ability (Fig. 8d). A compression test was conducted on the pyramid-unit trusses and the force-displacement curve results presented in Fig. 8b, indicating that the structure can withstand up to 2805 times its own weight before being damaged. It can also be found in the figure that when the fibers are off-center distributed on the upper surface of the composite material, the mechanical bearing capacities of the truss structures are higher than when the fibers are off-center distributed on the side surface, which is consistent with the simulation analysis results, reflecting the impact of fiber distribution on the mechanical properties of the CFRLCE structures. In addition, a continuous fibers/LCE structure can be used to realize the integration of the deformation ability and bearing capacity. As shown in Fig. 8e, twelve filaments with embedded fibers off-center were prepared using the CFDIW 4D printing method and built into a cubic truss structure. When the light curing time was 1 h, the modulus of the structure was low and could not even support a coin. However, it could produce evident shrinkage deformation at a heating temperature of 130 °C, as shown in Fig. 8g and j. Subsequently, the structure was irradiated with ultraviolet light until the total light curing time was 12 h. Further, the molecular crosslinking degree was improved, which enhanced the stiffness of the structure and allowed it to carry a weight of 20 g with a self weight of 0.225 g. Conversely, its molecular crosslinking degree was not high enough to seriously damage its deformation capacity. Therefore, the truss could simultaneously maintain the mechanical bearing characteristics and deformation function (Fig. 8f and i), indicating the versatility of the CFDIW 4D printed CFRLCE structures.

### Discussion

In summary, we have demonstrated a CFDIW 4D printing process for CFRLCE, which can directly print three-dimensional structures with the support of continuous fibers. The off-center position of the continuous fibers in the core-shell composite structure can be directly controlled, resulting in programmable deformation with high controllability. In this study, we realized the adjustable design of large deformation and high bearing capacity of LCE, which can carry a load of up to 2805 times its own weight, and achieved a bending deformation curvature of 0.33 mm$^{-1}$ at 150 °C.

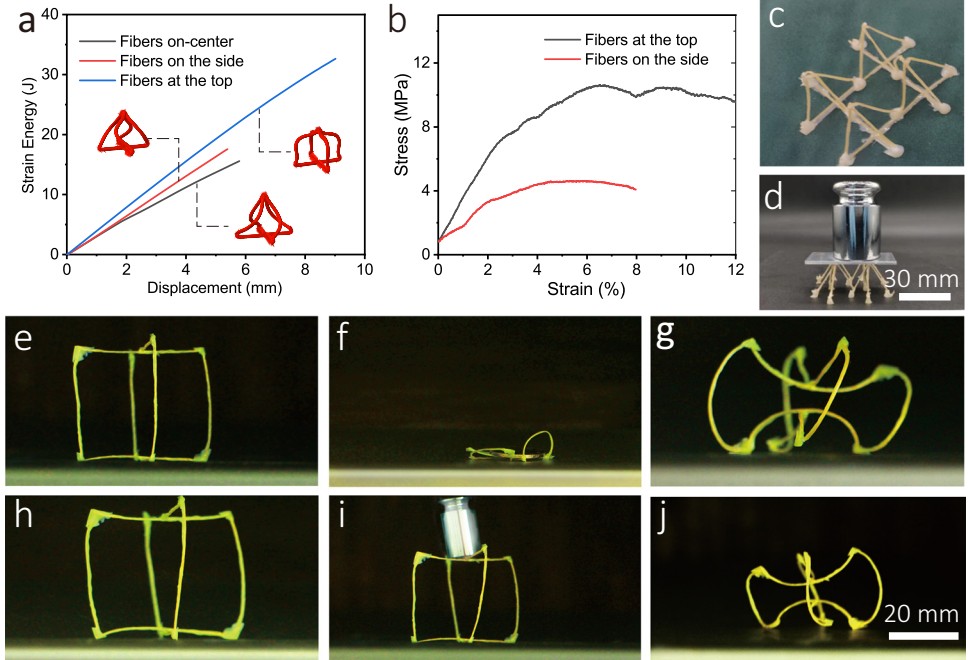

**Fig. 8 | Bearing capacity of the LCE trusses. a** Simulation of the compression process of truss structures with different distribution of fibers, illustrating a better bearing performance of the off-center composite. **b** Force-displacement curve for the pyramid-shaped truss, showing that truss structure that withstood up to 2805 times its own weight. **c** Truss structure for the four pyramid-shaped units. **d** The bearing effect of pyramid structure. **e** Cubic truss constrained using twelve CFDIW 4D printed filaments. **f** Bearing capacity of the cubic truss with a light curing time of 1 h. **g** Deformation ability of the cubic truss with a light curing time of 1 h. **h** Cubic truss with light curing time of 12 h. **i** Bearing capacity of the cubic truss with light curing time of 12 h. **j** Deformation ability of the cubic truss with a light curing time of 12 h.

Furthermore, high bearing and deformation capacities were simultaneously obtained in a CFRLCE truss through the design optimization of the structure. Due to its deformation ability and mechanical properties, the structures prepared by the CFDIW process are expected to be applied in the field of soft robotics with gripping functions, such as soft robotic arms or drug delivery robots.

Our future work is to explore and realize the actuating ability of CFRCLE composites, which is also the original goal of this paper. In this study, when the temperature is not high enough to cause significant deformation of the structure, the modulus of LCE decreases seriously. It leads to that although the structure has achieved bearing capacity and deformation capacity, the bearing capacity at higher temperature is not excellent. There are some methods may improve the actuating ability of composite materials, such as replacing liquid crystal materials with photoinduced deformation LCE. Some methods are expected to improve the actuating ability of the composites, such as replacing liquid crystal materials with photoinduced deformation LCE, or replacing fiber materials with continuous shape memory polymer fibers with deformation ability. Once the actuation capability is achieved, this research can be used to develop new avenues for creating soft robotics, mechanical metamaterials, and artificial muscles.

## Methods
### Continuous fibers
The continuous fiber material used in this work is aramid fiber (Sovetl, China), which has an axial CTE of $-2 \times 10^{-6}\,K^{-1}$, a tensile strength of 3.6 GPa and a tensile modulus of 131 GPa.

### LCE Ink
The liquid crystal ink was synthesized in one step in the ink chamber using a catalyst-free addition reaction. 1,4-bis(4-(6-(acryloxy)hexyloxy) benzoyloxy)−2-toluene (R6M, Acmec, 2.62 g), 2-Methyl-1,4-phenylene-bis(4(3(acryloyloxy)propyloxy)benzoate) (RM257, Acmec, 0.72 g), 1,4-benzenedimethylmercaptan (Yuanye, 0.44 g), Benzoin dimethyl ether or benzyl dimethyl ketal (DMPA, Yuanye, 0.12 g), and fluorescent agent 803 C (Pantone, 0.1 g) were added to the ink chamber (the molar ratio of R6M to RM257 is 1.0: 0.3). The mixture was heated at 160 °C for 30 min to allow the formation of polymer chains by the liquid crystal monomers upon the reaction of chain extenders, and the synthesized products are shown in Supplementary Fig. 1. Transfer the molten liquid crystal polymer to the chamber where continuous fibers have already penetrated, and slowly cool the liquid crystal to the printing temperature.

### 4D printing
The main difference between our CFDIW 4D printing equipment and existing DIW equipment involves the structure of the chamber with an off-center channel, the piston with a capillary channel, and the coupling. The heating rod and thermocouple were embedded in the chamber to heat the liquid crystal and maintain the printing temperature. First, the fiber was passed through a perforated piston, ink chamber, printing nozzle, and perforated platform. Subsequently, the powdered liquid crystal mixture was filled into the chamber and capillary hole, heated to 160 °C and stirred continuously for 30 min. During the 4D printing process, a higher curing strength is required at the filament ends. Therefore, a high-power point light source was used to irradiate it, while the rest of the filament is illuminated by an area light source. After 4D printing, the structure was put into an ultraviolet curing box to ensure uniform exposure.

### Wetting angle
The liquid crystal formula was heated at 160 °C for 30 min before being cooled to room temperature to simulate the temperature changes

during 4D printing. Subsequently, drops of molten liquid crystals were deposited onto a horizontally-straightened continuous aramid fiber surface, the interface between the two materials was photographed using a non-contact measuring instrument (Kruss, DSA100, Germany), and the wetting angle is calculated.

## DSC

The liquid crystal monomers were heated from room temperature to 150 °C at a rate of 10 °C min$^{-1}$. To ensure that each component reaches its clear point, we conducted DSC tests (TA, Discovery DSC250, USA) on the R6M, RM257, and DIW ink formulation respectively.

## Shear viscosity

0.3 g of liquid crystal formula is heated at 160 °C for 30 min, and then gradually cooled to simulate the temperature change during printing. The rheological properties were tested at different cooling stages. (HAAKE, Mars60, Germany).

## 2D XRD

To obtain the molecular chain orientation of liquid crystal components in composite trusses, the continuous fibers were stripped from the composites and ten 30 mm long LCE filaments were cut and bundled for XRD test (Rigaku, HomeLab, Japan). The anode target X-ray source used was a Cu K$\alpha$ radiation with a maximum output power of 2.97 kW and an electron beam focal spot diameter of 70 μm, and the detector used was a Hypix-6000 photon direct reading detector. The Hermans orientation parameter was calculated using the azimuthal integration from Fit2D software, and the following equation is used to calculate the orientation degree of the polymer:

$$P = 1 - \frac{3\int_0^{2\pi} I(\theta)[\sin^2\theta + \sin\theta\cos^2\theta \ln(1 + \tan\theta)]d\theta}{2\int_0^{2\pi} I(\theta)d\theta} \qquad (2)$$

where $I(\theta)$ represents the signal intensity in different directions on the XRD pattern.

## SEM

The composite filament was cut into sections, and the cross section is used to observe the off-center distribution and the impregnation effect of the fibers. The SEM images were taken by a electron microscope (Hitachi, SU3500, Japan) using a voltage of 5 kV. Before observation, the samples were sputtered with gold in vacuum for 120 s with a current of 40 mA to enhance the conductivity of their surfaces.

## Micro-droplet debonding test

The single aramid fiber was tightened and fixed on the sample holder, and micro-droplets were prepared by dipping the fiber in the melted liquid crystals. Micro-droplets were stuck on the crosshead of the interfacial evaluation equipment (Model, HM410, Japan), and were debonded from the single fiber by moving the sample holder with the speed of 0.12 mm s$^{-1}$.

## Simulation

The finite element analysis of the thermal-induced deformation process of continuous fibers reinforced LCE composites was performed by ABAQUS. The simulation included the parameters: the elastic modulus, the Poisson's ratio and the CTE. The LCE structure in the simulation were assumed to have great thermal shrinkage property, and the initial temperature was set to room temperature (25 °C). The composite truss was modeled as a cylinder, and the cross-section of reinforced continuous fiber bundle was modeled as a circle and an off-center bow, which were used to simulate the case of low and high off-center degree, respectively. The specific shape parameters were obtained from the SEM images shown in Fig. 3. Second-order C3D8I elements were used during the simulations. In the analysis of the

pyramid-shaped structure, initial geometric imperfections were introduced to the mesh to better trigger the torsional deformation behavior of the structure. For the simulation result figures in this article, the different colors represent the magnitude of displacement during the deformation process.

## Data availability

Source data are provided with this paper.

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

## Acknowledgements

This work was supported by the National Natural Science Foundation of China (52075422) to X.T. We also thank the State Key Laboratory for Manufacturing Systems Engineering and the Instrumental Analysis Center of Xi'an Jiaotong University for facility support.

## Author contributions

Experiments were designed by Q.W., X.T., and D.L., and conducted by X.T. and D.L. The 4D printer was developed by Q.W. The deformation curvature was measured by D.Z. The LCE material was synthesized by Y.Z. The mechanical models and finite element calculations were developed by W.Y. The paper was written by Q.W. and revised by X.T. All authors discussed the results and their implications.

## Competing interests

The authors declare no competing interests.
