## [Peer Review File · Nature Communications]

Programmable spatial deformation by controllable off-center freestanding 4D printing of continuous fiber reinforced liquid crystal elastomer compositesREVIEWER COMMENTS

Reviewer #1 (Remarks to the Author):

Subject: Reviewer Recommendation and Comments for Manuscript Number NCOMMS-22-42090 submission to Nature Communications

General comment:

This manuscript presents an approach to directly print 3D structures with the support of continuous fibers. The authors use the off-center position of the continuous fibers in the core-shell composite structure. Although the content of the paper will be quite interesting for the community, some important studies related the scope of this paper are not presented, which may affect my recommendation for publication.

Overall, the paper is fair written and easy to read. Nevertheless, I do have several questions and comments for improvement to recommend publication.

Specific comment:

Comment 1:

The presentation of the paper needs to be improved to be clear and consistent, like:

Abstract

Keywords

Introduction

Materials and method

Results and discussion

Conclusion

References

Comment 2:

Suong Hoa has great effort in 4D printing using of continuous fibers. For example:

S. Van Hoa, Factors affecting the properties of composites made by 4D printing (moldless composites, *Adv. Manuf. Polym. Compos. Sci.* 3 (2017) 101–109.

<https://doi.org/10.1080/20550340.2017.1355519>.

S. Van Hoa, Development of composite springs using 4D printing method, *Compos. Struct.* 210 (2019) 869–876. <https://doi.org/10.1016/j.compstruct.2018.12.003>.

S. V Hoa, X. Cai, 2020. Twisted composite structures made by 4D printing method, *Compos. Struct.* 238, 111883. <https://doi.org/10.1016/j.compstruct.2020.111883>.

Why these related papers are not added?

Comment 3:

Line 8-9, page 1

“Majority of the existing 4D printing methods can only fabricate planar structures, which limits their deformation designability and bearing capacity”. This is a loose statement, please be specific.

Comment 4

Line 13, page 1

“significantly improved by the continuous fibers”. Is there any quantitative percentage?

Comment 5

Line 24, page 2,

Please make introduction heading

Comment 6
Lines 34-37, page 2

"LCEs are composed of an orientated liquid crystalline polymer that has a high thermal shrinkage effect along the direction of the molecular chain and the orientation of the LCE molecular chain determines the mechanical property of the polymer". Please reworded this statement.

Comment 7
Line 55-56, page 3

Please make a comparison between the deformation mode presented in this paper and the ones presented in Hoa's studies and mention your contributions

Comment 8
Lines 64-65, page 3

"For these reasons, most 4D printing methods for LCE structures or bilayer composites are either planar structures or simple shapes obtained by folding, curling ...". This seems not true, please verify.

Comment 9
Lines 81-84, page 4

"Considering LCE as an example, most of the existing 4D printed LCE structures are linear or planar and can only produce tensile stress during shrinkage deformation, which prevents them from maintaining their structural stability under the external compression forces". What about different materials, please highlight the novelty in terms of processing and materials' selection.

Comment 10
Lines 89, page 5

"a new type of 4D printed composite structure is required." It is existing.

Comment 11
Lines 111, page 6

Method section should be presented before the results.

Comment 12
Lines 133, page 7

"room temperature was 32.1°". Please check.

Comment 13
Lines 141-142, page 7

How you control the impregnation level? This information must be provided.

Comment 14
Lines 161, page 8

Please provide more details about ABAQUS model.

Comment 15
Lines 166, page 8

Controlling the distribution of the fibers is a challenging process. How you control that?.

Comment 16
Lines 206-208, page 10

The statement is not clear, please reword it.

Comment 17

Lines 244, page 12

Is the impregnation level affect the deformation curvature as well?.

Comment 18

Lines 253, page 12

"..the better is the deformation performance of the composite structure". Do think this is related to the degree of curing?

Comment 19

Lines 290-292, page 14

Rewrite is required.

Comment 20

Lines 327, page 15

Please avoid the repetition in discussion part. Also. The discussion part seems poor.

Comments for Manuscript #NCOMMS-22-42090

The authors studied the influence of (spatial) 3D printing parameters on liquid crystal elastomer encompassing continuous Fiber. Here they claim that both the printing speed and light source for curing have an outstanding role in spatial 3D printed samples; also, the composite containing off-center Fiber shows better results regarding mechanical features. There is no clear-cut hypothesis or deep discussion in the current version of the paper, especially for chemical analysis and simulation parts. Despite that, this manuscript's novelty and originality are a little above-average, but it needs major modification to be ready for publishing in Nature Communications.

1) Please be more specific in your title (it is charming now but not clear): I think it is not 4D printing; just replace 3D printing.

For instance,

Understanding Spatial Deformation by Controllable Off-center Continuous Fiber 3D Printing

2) There are some typo mistakes in the manuscript. Please take care of them, like Page 12, lines 251 and 252, ... *The molecular orientation has a significant influence on the molecular chain orientation????...*

3) Material selection:

What is the material of Fiber? It is not clear at all.

What is the rationale behind the use of this LCE instead of a well-known matrix?

4) You should mention the mechanism of mechanical strengthening and present a convincing description of "why fiber is able to be wet by LCE?". In the current version, the authors merely reported the numbers and observations. So please evaluate the chemical analysis of Fiber and understand the role of interaction with the matrix.

4) As clear as crystal, the continued Fiber can improve the mechanical properties. However, you just study static mechanical properties regardless of dynamic properties: Is it true?. Please imply dynamic phenomenon with the aid of conducting DMTA test in the manuscript and provide reasonable reasons for the mechanical improvement and chain orientation. *For instance*, the polymer chain moves more easily in XX sample because of the printing angle and free volume and slippage on the surface of Fiber that is able to enhance the chance of fiber pull-out.

5) Is it either a 3D printed sample or a 4D printed (read about them)?

One of the most effective parameters in your sample is "Programmable Spatial Deformation", which is overlooked in this study. What is the meaning of programmable? I could not find any 4D printed sample that has the capability of changing its shape upon stimuli field. Deformation of samples due to curing time is not controllable and acceptable as a 4D printing.

6) In the majority of figures and results, simulation is spotlighted. I suppose you can find more details of the simulation part in ABACUS; see other papers.

7) Due to lacking controversial debits in the curing part, I would postpone the in-depth science-oriented discussion to the revision stage. Only one-point springs to my mind: you tried to cure it and connect all

the macromolecules. Please make sure that outstanding mechanical behavior comes from which source: curing or Fiber.

Reviewer #3 (Remarks to the Author):

This manuscript presents a new approach to printing temperature-responsive LCE composites consisting of a soft LCE matrix as a shell and rigid aramid fiber bundles as a core. Intriguingly, the authors developed a customized direct-ink-writing apparatus that can simultaneously print the aramid fiber cores and LCE shell. During the printing process, the aramid fibers were intentionally distributed off-center within the composite fibers to induce bending deformation based on the mismatch of CTE values between the LCE matrix and aramid fiber. The author claimed that the location of aramid fibers can be precisely controlled to programmable bending. While the concept of off-center continuous LCE composite fibers is quite unique and interesting, the current manuscript is not technically sound because the fundamental deformation mechanism of the fiber as well as the printing process is not clear. Also, the quality of the manuscript including figures/figure captions and details of explanation is not satisfactory. Therefore, I recommend the authors should refine the manuscript, and submit it elsewhere. The following comments could improve the quality of this manuscript.

- 1) Page 4: the authors claim that one of the challenges of 4D printed structure is poor mechanical bearing capacity caused by the material and shape restrictions. However, some of the examples listed in page 4 is not so related mechanical bearing capacity of 4D printed materials. The logic of this paragraph should be revisited.
- 2) Can the authors provide any results of the interfacial adhesion between LCE and aramid fiber?
- 3) There is no information on aramid fiber such as chemical structure, molecular weight, vendors, etc.
- 4) It would be much helpful if the authors can provide videos that describe the printing process and thermal actuation of each printed structure.
- 5) The simulation result of Fig 2d does not show much detail and the importance of the result. Also, there is no information regarding the color distribution in Fig 2d.
- 6) What is the "m" in Fig 1a? Please provide details because the authors seem to use a mixture of R6M and RM257. What is the molar ratio of the LC monomer? Also, what is the molar ratio between the LC monomers and the thiol-chain extender? – this is a critical molecular parameter to determine materials' properties. Also, the diacrylate-terminated LC oligomer structure in Fig 1a is not correct – please check out the reaction details for this chemistry reported by T. White (Macromolecules 2021, 54, 23, 11074–11082)
- 7) Please redraw of Fig 1c with x-axis with log-scale. Many 3D or 4D printed LCE papers used the log scale for both x- and y-axes.
- 8) The theta, alpha and beta are not presented in Fig 3. Please include this information in Fig 3.
- 9) SEM images in Fig 3g do not clearly support off-center positioned aramid fiber. The resolution should be much improved, and higher magnification images should be provided as well.
- 10) The raw data for 2D-XRD images should be provided, especially azimuthal plots.
- 11) The experimental conditions for XRD and SEM are not provided.
- 12) What is the reason for using a fluorescent agent?
- 13) In Fig 2, the authors show the schematic for the printing apparatus. But I think more details should be provided for each part. The printing procedure presented in the manuscript and Fig 2 is not enough to completely understand the exact principle of printing process and the way to control the location of the aramid fiber core.
- 14) Fundamental deformation mechanism of the composite fiber should be more clearly presented. I do not think this was not thoroughly explained in the paragraph or in the figure. This is important because this will eventually explain the deformation process of other complex printed structures.

Reviewer #4 (Remarks to the Author):

Some recent papers have reported the 4D printed liquid crystal elastomer using the direct ink writing method. However, this work introduced continuous fibers in the liquid crystal elastomer using a direct ink writing method for 4D printed spatial structures with good mechanical properties. Although the paper contains interesting work, some major questions need to be answered. Here are specific points for consideration:

1. Why are the fibers beneficial to support the direct manufacture of spatial 3D structures? What is

the mechanism? What is the difference with the recent publications on self-supporting, for example, DOI: 10.1002/adma.202204890?

2. The mechanical properties of the polymer matrix can be certainly improved with the continuous fibers, such as the traditional carbon fiber reinforced polymer matrix laminates and the 3D printed continuous fibers reinforced polymer-matrix composites. This work also utilized continuous fibers to improve the mechanical properties of the liquid crystal elastomer, however, the off-center distribution of continuous fibers is harmful to the mechanical properties, how to avoid it and make the on-center distribution?

3. The potential applications and prospects of 4D printed continuous fiber-reinforced liquid crystal elastomer composites are unclear. Please clarify them.

4. In Fig. 7, the 4D printed truss in this work could withstand up to 2805 times its own weight. It is not a surprising result, and 4D printed truss with many continuous fibers reinforced polymers can also realize a similar result. Besides, the bearing capacity is not the important research point for 4D printed liquid crystal elastomer (active material), and the authors should give a surprising actuation capability result of 4D printed liquid crystal elastomer instead of the bearing capacity.

5. The model-to-part fidelity of 4D printed continuous fiber reinforced liquid crystal elastomer (Fig. 6-7) in this work seems not good.

6. The organization of figures in the manuscript is not good.

Response to Reviewers

The following is a point-to-point response to the reviewers' comments.

Reviewer #1

Subject: Reviewer Recommendation and Comments for Manuscript Number NCOMMS-22-42090 submission to Nature Communications

General comment:

This manuscript presents an approach to directly print 3D structures with the support of continuous fibers. The authors use the off-center position of the continuous fibers in the core-shell composite structure. Although the content of the paper will be quite interesting for the community, some important studies related the scope of this paper are not presented, which may affect my recommendation for publication.

Overall, the paper is fair written and easy to read. Nevertheless, I do have several questions and comments for improvement to recommend publication.

Answer: Thank you for the comments on the paper. We have added some experiments and simulations to analyze the mechanism of CFDIW 4D printing. We have revised the manuscript and redrew the figures as suggested. The specific modifications are as follows:

I) The mechanism of interfacial properties and mechanical performances of LCE composites have been analyzed in greater depth, and it has been proved that off-center distribution of fibers has an improvement on interfacial properties and some mechanical performances.

II) SEM observation experiment for impregnation effect and FTIR experiment for verifying reaction mechanism have been supplemented.

III) The simulations of thermal deformation and compression behavior of three-dimensional structures have been supplemented, and the simulation model has been optimized.

IV) Major revisions have been made to the manuscript to supplement the introduction of the CFDIW process mechanism and some technical details.

V) Videos of 4D printing process and deformation process have been added, and photos of deformation effects and fiber off-center effects have been added.

VI) All the figures have been modified.

In particular, we would like to thank you for your suggestions on the evaluation of the impregnation effect. Therefore, we found that the impregnation effect of the composite can be changed by adjusting the off-center degree of the fiber bundle, and the impregnation effect can be improved by increasing the off-center degree. In the unrevised manuscript, we only focus on the influence of off-center fiber distribution on deformation performance and bearing capacity.

Specific comment:

Comment 1:

The presentation of the paper needs to be improved to be clear and consistent, like:

Abstract

Keywords

Introduction

Materials and method

Results and discussion

Conclusion

References

Answer: Your suggestion is greatly appreciated. We have adjusted the structure of the paper, and now the article is organized in the following order:

Abstract

Keywords

Introduction

Results (Including: CFDIW 4D printing method for off-center CFRLCEs, Controlling the off-center degree of the fibers, Controlling the off-center direction of the fibers, Deformation properties of CFRLCEs, Controllable deformation ability of 4D printed trusses and Bearing capacity of 4D printed trusses)

Discussion

Methods

Data availability

References

Competing interests

According to the requirements of *Brief guide for submission to Nature Communications*, methods section is presented behind results section. Nevertheless, we have revised the results section, and our 4D printing materials and methods is briefly introduced at the beginning of *Results*.

Comment 2:

Suong Hoa has great effort in 4D printing using of continuous fibers. For example:

S. Van Hoa, Factors affecting the properties of composites made by 4D printing (moldless composites, *Adv. Manuf. Polym. Compos. Sci.* 3 (2017) 101 – 109. <https://doi.org/10.1080/20550340.2017.1355519>.

S. Van Hoa, Development of composite springs using 4D printing method, *Compos. Struct.* 210 (2019) 869 – 876. <https://doi.org/10.1016/j.compstruct.2018.12.003>.

S. V Hoa, X. Cai, 2020. Twisted composite structures made by 4D printing method, *Compos. Struct.* 238, 111883. <https://doi.org/10.1016/j.compstruct.2020.111883>.

Why these related papers are not added?

Answer: Your suggestion is greatly appreciated. We have studied these related papers. We quoted the work of Hoa et al. and compared it with our research. We added the following sentence to the article:

“Hoa et al. prepared laminates composed of materials with different thermal expansion coefficients by 4D printing to realize the moldless composites manufacturing.”

These related works have been cited as follows:

“36 Hoa, S.. Factors affecting the properties of composites made by 4D printing (moldless composites manufacturing). *Adv. Manuf. Polym. Compos. Sci.* 3 101 (2017).

37 Hoa, S.. Development of composite springs using 4D printing method. *Compos. Struct.* **210** 869 (2019).

38 Hoa S., and Cai X.. Twisted composite structures made by 4D printing method. *Compos. Struct.* **238** 111883 (2020).”

Comment 3:

Line 8-9, page 1

“Majority of the existing 4D printing methods can only fabricate planar structures, which limits their deformation designability and bearing capacity” . This is a loose statement, please be specific.

Answer: Your suggestion is greatly appreciated. We are sorry that the statement is not precise enough. Some 4D printing methods can prepare freestanding structures. However, 4D printing works for liquid crystal materials is difficult to prepare freestanding structures [1, 2]. We have revised this statement as follows:

“However, majority of the existing 4D printing methods for liquid crystal elastomers can only fabricate planar structures, which limits their deformation designability and bearing capacity.”

Comment 4

Line 13, page 1

“significantly improved by the continuous fibers” . Is there any quantitative percentage?

Answer: In order to analyze the improvement of mechanical properties of fiber materials, we tested the tensile properties of liquid crystal materials without continuous fibers. When the light curing time is 10 min, the average tensile modulus of pure LCE material is 0.24 GPa. In comparison, the tensile modulus of continuous fiber reinforced LCE is 2.8 GPa, with an increase of about 10.7 times.

In this paper, we introduced the improvement of fiber on the mechanical properties of LCE, as follows:

"In comparison, the tensile modulus of LCE material without continuous fibers is only 0.24 GPa when the light curing time is 10 min, indicating the improvement of the mechanical property by continuous fibers."

Comment 5

Line 24, page 2,

Please make introduction heading

Answer: Thank you very much for this greatly appreciated suggestion. We have made the heading of Introduction.

Comment 6

Lines 34-37, page 2

“LCEs are composed of an orientated liquid crystalline polymer that has a high thermal shrinkage effect along the direction of the molecular chain and the orientation of the LCE

molecular chain determines the mechanical property of the polymer” . Please reworded this statement.

Answer: Thank you very much for this greatly appreciated suggestion. We have reworded this statement, and the revised sentence is as follows:

“LCEs with uniformly oriented polymers have obvious anisotropy, which shows the thermal shrinkage effect and better mechanical properties along the molecular chain orientation.”

Comment 7

Line 55-56, page 3

Please make a comparison between the deformation mode presented in this paper and the ones presented in Hoa’s studies and mention your contributions

Answer: We quoted the work of Hoa et al. and compared it with our research. We added the following sentence to the article:

“Hoa et al. prepared laminates composed of materials with different thermal expansion coefficients by 4D printing to realize the moldless composites manufacturing.”

These related works have been cited as follows:

“36 Hoa, S.. Factors affecting the properties of composites made by 4D printing (moldless composites manufacturing). *Adv. Manuf. Polym. Compos. Sci.* **3** 101 (2017).

37 Hoa, S.. Development of composite springs using 4D printing method. *Compos. Struct.* **210** 869 (2019).

38 Hoa S., and Cai X.. Twisted composite structures made by 4D printing method. *Compos. Struct.* **238** 111883 (2020).”

Comment 8

Lines 64-65, page 3

“For these reasons, most 4D printing methods for LCE structures or bilayer composites are either planar structures or simple shapes obtained by folding, curling ...” . This seems not true, please verify.

Answer: Your suggestion is greatly appreciated. At present, most of the structures prepared by LCE-based 4D printing process are planar structures [3-5], and some important works on the 4D printing methods for 3D LCE structures [1, 2] have been cited in our manuscript. The reason why the liquid crystal is rarely printed into a 3D structure is that the orientation of the liquid crystal molecular chain is along the motion path of the nozzle, while most 3D printing is layer-by-layer printing, which cannot make the liquid crystal orientation in the z direction, so it is difficult to produce a 3D deformable structure. Guo et al. [1] discussed why it is difficult for LCE to prepare 3D structure as follows:

“Additionally, the printed LCE director field is along with the extruded fiber and cannot change abruptly without changing the printing path. At last, the current extrudable LCE materials do not have a high enough elastic modulus to sustain a printing path along the z direction.”

“However, the fabrication of LCEs with both arbitrary 3D geometries and arbitrary 3D director fields is still challenging and in high demand for enriching desirable 3D-to-3D morphing modes towards real-world applications.”

The solution of this work is based on the predefined director field. The orientation of

molecular chains need to be adjusted voxel-by-voxel during 4D printing, which leads to low manufacturing efficiency. In addition, the orientation of each voxel can only follow several specific directions.

The preparation of freestanding LCE structure is a feasible solution to this problem, because it allows the liquid crystal to orient along any direction, including the z direction. However, a challenge of printing freestanding liquid crystal truss structure is that liquid crystal materials are easy to hang down while curing. This is one of the reasons why we introduce continuous fibers into the LCE structure to prepare composites.

However, a paper entitled *4D Printing of Freestanding Liquid Crystal Elastomers via Hybrid Additive Manufacturing* [2] was published on *Advanced Materials* in August 2022, which realized the freestanding of liquid crystal materials. It uses multiple high-power lasers to aim at the exit direction of printing, which greatly reduces the curing time of liquid crystal. Peng et al. [2] discusses why it is difficult for LCE to prepare three-dimensional structure as follows:

“Although 3D structures, such as pinecone and saddle-shaped structures, can be achieved by 2D structures via different actuation strains between layers, the layer-by-layer manner of material deposition in DIW makes LCEs to be printed on the build platform or the previous layers. As a result, the actuation of the printed LCE structures is limited to planar shrinkage, simple bending, or twisting.”

“Still, this approach only aligns mesogens in one direction; hence, applications are limited to thin films and simple bending actuation.”

Our manuscript was submitted in October 2022, less than two months later than the publication of this paper, and we did not notice and cite this work at that time. Now the revised version have cited this study and made full comparison with our work.

Although their work [2] and our work have both realized the 4D printing for freestanding LCE, there are still many differences between us: i) In their work, additional support structures need to be provided to fix both ends of the liquid crystal filaments, and different 4D printed structures need to rely on different support structures, while our 4D printing process does not need support structures; ii) Continuous fiber is introduced into our 4D printing materials, which makes the structure have better bearing capacity; iii) Our work does not require expensive multiple laser sources, thus reducing the printing cost.

In our revised manuscript, the progress of 4D printing of the three-dimensional LCE structure is re-introduced as follows:

“Peng et al. realized the 4D printing of the freestanding LCE structure by using multiple laser sources to cure the LCE in-situ when the nozzle moves and extrudes the liquid crystal material. However, this method has the following limitations: i) the forming process of LCE structures depends on Supplement structures; ii) The LCE truss structure has little bearing capacity; iii) The solidification of liquid crystal materials requires expensive high-power multiple laser sources.”

Comment 9

Lines 81-84, page 4

“Considering LCE as an example, most of the existing 4D printed LCE structures are linear or planar and can only produce tensile stress during shrinkage deformation, which prevents them from maintaining their structural stability under the external compression forces”. What about

different materials, please highlight the novelty in terms of processing and materials' selection.

Answer: Thank you for the comments on the paper. In fact, there are many kinds of 4D printing matrices that we have considered, such as thermoplastic resins, shape memory polymers (SMPs), and hydrogels.

Our previous works are related to the 4D printing for composites with continuous fiber reinforced thermoplastic matrix, and the matrix material with good deformation effect is PA-based composite. However, its deformation ability is far inferior to that of LCE matrix composites. When the temperature is heated to 150 °C, the curvature change of the continuous fibers reinforced PA composite is about 0.028 mm^{-1} , while the curvature change of the continuous fibers reinforced LCE composite used in this paper is about 0.33 mm^{-1} .

We also considered SMPs as matrix materials due to their excellent mechanical properties. However, their deformation process is not continuous. SMPs can only change the status directly from temporary shape to final shape, and the deformation process is usually irreversible. [6, 7]

Hydrogels are also often used as 4D printing materials due to their excellent deformability [8], and have been studied in our previous exploration. However, our experiments showed that the interface bonding effect between the hydrogel and the continuous fiber is not good because the surface of the hydrogel is very wet.

LCE material has advantages in both thermal deformation ability and mechanical property [9], so it is selected as the matrix for 4D printing in our research, even though the 4D printing for freestanding continuous fibers reinforced liquid crystal materials is a great challenge.

We have added the consideration of material selection into the introduction part of this paper, as follows:

“LCEs are smart materials with high deformation ability, relatively significant mechanical performance, rapid actuation and reversible deformation process that can be prepared by direct ink writing (DIW) 4D printing, and can respond to various external stimuli, such as light, heat, and electric fields.”

“As a material with programmable molecular chain orientation, LCE is expected to produce the freestanding, bearable and deformable three-dimensional structure.”

Comment 10

Lines 89, page 5

“a new type of 4D printed composite structure is required.” It is existing.

Answer: We are sorry that this statement is not accurate enough. Although the preparation of freestanding 3D or 4D printing structures using pure liquid crystals [2] or continuous fibers reinforced thermoplastic resins [10] have existed, the DIW 4D printing for continuous fiber reinforced liquid crystal materials has not appeared in previous works we have learned. In order to avoid ambiguity, we have revised the statement:

“To solve these problems, a 4D printing method for freestanding LCE composite truss structure is required. The high bearing capacity of the 3D structure comes from the mechanical reinforcement in three directions, and the deformation controllability is based on the bendable trusses.”

Comment 11

Lines 111, page 6

Method section should be presented before the results.

Answer: Thank you for the comments on the paper. We have adjusted the structure of the paper, and now the article is organized in the following order:

“Abstract

Keywords

Introduction

Results (Including: CFDIW 4D printing method for off-center CFRLCEs, Controlling the off-center degree of the fibers, Controlling the off-center direction of the fibers, Deformation properties of CFRLCEs, Controllable deformation ability of 4D printed trusses and Bearing capacity of 4D printed trusses)

Discussion

Methods

Data availability

References

Competing interests”

According to the requirements of *Brief guide for submission to Nature Communications*, methods section is presented behind results section. Nevertheless, we briefly introduced our 4D printing materials and methods at the beginning of *Results*.

Comment 12

Lines 133, page 7

“room temperature was 32.1° ” . Please check.

Answer: We are sorry for the ambiguity in our writing. Here, “32.1°” refers to the wetting angle rather than the room temperature. In order to avoid ambiguity, we revised this sentence:

“The average wetting angle of the liquid crystals and aramid fibers was 32.1° at room temperature, as shown in Figure S6, indicating that liquid crystals have the ability to spread on the surface of the fiber bundle.”

Comment 13

Lines 141-142, page 7

How you control the impregnation level? This information must be provided.

Answer: Your suggestion is greatly appreciated. In order to verify the impregnation effect of the composites, we observed the SEM images of the composite cross section with higher magnification, as shown in the following figure. These new SEM results were added to the article and *Supplement Information*.

Figure 1 SEM images of the cross section of the composite with good impregnation effect

Figure 2 SEM images of the cross section of the composite with poor impregnation effect

As shown in Figure 2, when the off-center degree of the fiber is low during the 4D printing process, the cross section of the fiber bundle is approximately circular, and the liquid crystal is difficult to immerse into the interior of the fiber bundle. The structures with different printing speeds were observed by SEM, and the impregnation effect could not be improved. However, when we reduce the inclination angle in the printing process to -10° , the impregnation effect of the material is significantly improved. The SEM image of the cross section of the prepared composite structure is shown in Figure 1. It can be seen that the cross section of the fiber bundle becomes flat, which makes it easier for the liquid crystal to immerse into the interior of the fiber bundle, and no obvious holes can be observed in the SEM image.

In summary, the impregnation effect can be controlled by adjusting the degree of fiber off-center degree. We added some descriptions in the manuscript as follows:

“In the process of composite forming, the off-center distribution is helpful for the liquid crystal to impregnate on the fiber surface. When the lateral force of the nozzle is low during the pulling process, the cross section of the fiber is close to the circle, and there are some cavities not filled with liquid crystal inside (Fig. S5). The higher off-center effect makes the cross-section shape of the fibers more flat, so that the liquid crystal is easier to penetrate into the interior of the fiber bundle. The cross section of the pulled composite with a high off-center effect was observed by SEM, as shown in Fig. 1e, indicating that the impregnation of the two materials was sufficient.”

Lines 161, page 8

Please provide more details about ABAQUS model.

Answer: Thank you very much for this greatly appreciated suggestion. We added details of simulation analysis in the Methods section:

“Simulation. The finite element analysis of the thermo-induced deformation process of continuous fibers reinforced LCE composites was performed by ABAQUS. The simulation included the parameters: the elastic modulus, the Poisson’s ratio and the CTE. The LCE structure in the simulation were assumed to have great thermal shrinkage property, and the initial temperature was set to room temperature (25 °C). The composite truss was modeled as a cylinder, and the cross-section of reinforced continuous fiber bundle was modeled as a circle and an off-center bow, which were used to simulate the case of low and high off-center degree, respectively. The specific shape parameters were obtained from the SEM images shown in Figure 3. Second-order C3D8I elements were used during the simulations, and the convergence was examined. In the analysis of the pyramid-shaped structure, initial geometric imperfections were introduced to the mesh to better trigger the torsional deformation behavior of the structure.”

In addition, we optimized the simulation analysis and adopted a model with higher accuracy. In the previous simulation analysis, cylindrical liquid crystal material and elliptical continuous fiber material were used to model a single wire, while the model of truss structure was approximate. In the simulation analysis of the revised manuscript, the elliptical cylinder model with higher accuracy was adopted for all the structures, and the thermal deformation simulation analysis of the pyramid-shaped structure has been added.

Comment 15

Lines 166, page 8

Controlling the distribution of the fibers is a challenging process. How you control that?.

Answer:

When the fiber is pulled out of the nozzle, it is subjected to the shear force provided by the nozzle and thus is off-center distributed in the composite. In order to show how to control the off-center distribution of the fiber more clearly, a new figure was added to the article, and an introduction to the 4D printing mechanism of CFDIW process has been added to the manuscript:

“In order to prepare CFRLCEs, the continuous fiber bundle needs to be fully impregnated by liquid crystal polymer and adjusted to the off-center position of the composite filament, and then the composite is cured under ultraviolet light. Fig. 1c shows the CFDIW process in which the composite goes through three stages of impregnation, extrusion and curing. The fiber contacts with the liquid crystal in the chamber, and the liquid crystal is extruded out of the chamber while the fiber is pulled out of the nozzle to obtain a uniform CFRLCE composite. Since both ends of the fiber bundle are always tensioned, the fiber is subjected to the lateral force provided by the nozzle and thus leads to the off-center distribution of fibers in the composite filament. The cross section of the filament observed by scanning electron microscope (SEM) is shown in Fig. 1d, which proved the existence of the off-center distribution effect. In the process of composite forming, the off-center distribution is helpful for the liquid crystal to impregnate on the fiber surface. When the lateral force of the nozzle is low during the pulling process, the cross section of the fiber is close to the circle, and there are some cavities not filled with liquid crystal inside (Fig. S5). The higher off-center effect makes the cross-sectional shape of the fibers more flat, so that the liquid crystal is easier to penetrate into the interior of the fiber bundle. The cross section of the pulled composite with a high off-center effect was observed by SEM, as shown in Fig. 1e, indicating that the impregnation of the two materials was sufficient. In addition, it is necessary to prevent the liquid crystal from accumulating droplets on the surface of the fiber bundle to obtain the filament with the uniform shape during the curing process. The average wetting angle of the liquid crystals and continuous aramid fibers is 32.1° at room temperature, as shown in Fig. S6, indicating that liquid crystals have the ability to spread on the surface of the fiber bundle. In order to achieve good thermal deformation effect of the composite filament, the molecular chain of the extruded liquid crystal requires uniform orientation. The LCEs of the pulled filaments were tested

by 2D X-ray diffraction (XRD), as shown in Fig S7, indicating that the liquid crystal polymer chains are oriented. Owing to the difference of CTEs between fibers and liquid crystals, composites will produce bending deformation when heated.”

Comment 16

Lines 206-208, page 10

The statement is not clear, please reword it.

Answer: Your suggestion is greatly appreciated. We revised this statement as follows:

“Subsequently, the cross-sectional shape of the fiber bundle, which was later proved to be an important factor affecting the thermal deformation ability of composites, is affected by the lateral force on the fiber bundle.”

Comment 17

Lines 244, page 12

Is the impregnation level affect the deformation curvature as well?.

Answer: Your suggestion is greatly appreciated. In order to verify the impregnation effect of the composites, we observed the SEM images of the composite cross section with higher magnification, as shown in the following figure. These new SEM results were added to the article and *Supplement Information*.

Figure 1 SEM images of the cross section of the composite with good impregnation effect

Figure 2 SEM images of the cross section of the composite with poor impregnation effect

The impregnation effect can be controlled by adjusting the degree of fiber off-center degree,

and affect the deformation ability.

As shown in Figure 2, when the off-center degree of the fiber is low during the 4D printing process, the cross section of the fiber bundle is approximately circular, and the liquid crystal is difficult to immerse into the interior of the fiber bundle. The structures with different printing speeds were observed by SEM, and the impregnation effect could not be improved. However, when we reduce the inclination angle in the printing process to -10° , the impregnation effect of the material is significantly improved. The SEM image of the cross section of the prepared composite structure is shown in Figure 1. It can be seen that the cross section of the fiber bundle becomes flat, which makes it easier for the liquid crystal to immerse into the interior of the fiber bundle, and no obvious holes can be observed in the SEM image.

The experimental results show that the deformation curvature of the composite increases by an order of magnitude with the inclination angle decreasing from 90° to -10° . We speculate that the improvement of the interfacial adhesion between fiber and liquid crystal is one of the factors that improve the deformation ability of the composites. This is because the bending deformation mechanism of the composite is that the thermal expansion coefficients of the fiber and the liquid crystal are very different, so the stress will be generated between the two materials when heated, and the better impregnation effect is conducive to the stress transfer between the interfaces.

We added some descriptions in the article as follows:

“In addition, with the increase of off-center degree of the fibers, the cross section of the fibers with larger aspect ratio makes the cavities inside the fiber bundle disappear and leads to better impregnation effect. This will bring better deformation ability, because the bending deformation comes from the stress between the fiber and the liquid crystal with different CTEs, and a better interface bonding characteristic is obviously conducive to the transfer of stress between the two materials.”

Comment 18

Lines 253, page 12

“..the better is the deformation performance of the composite structure” . Do think this is related to the degree of curing?

Answer: Thank you very much for this greatly appreciated suggestion. We consider that the degree of curing may not be the main factor affecting the deformation performance. During the experiments with different extrusion speeds, the printed samples experienced the same curing time (5 minutes or 10 minutes) and the same UV intensity. In order to avoid the interference caused by the printing time, we only print some 30 mm long sample, and the printing time of different samples is less than 5 seconds. This difference is ignored compared to the curing time.

On the contrary, XRD experiments show that the difference of molecular chain orientation is very obvious, so we think that the difference of deformation properties is probably due to the degree of molecular chain orientation.

We have added the following statement to the article:

“In addition, the effect of printing speed on the curing time of liquid crystal can also be ignored.”

Comment 19

Lines 290-292, page 14

Rewrite is required.

Answer: Your suggestion is greatly appreciated. We rewrote this sentence as follows:

“Therefore, even for 4D printing structures with the same shape parameters, the deformation processes of the structures can be adjusted by the off-center positions of the continuous fibers, indicating the high controllability of the CFDIW process.”

Comment 20

Lines 327, page 15

Please avoid the repetition in discussion part. Also. The discussion part seems poor.

Answer: Your comments have been of great help to us. The discussion part of the submitted manuscript only summarizes the research results of this paper, without further prospects. The revised manuscript discusses the improvement direction and application prospect of this research, and specifically added the following contents:

“Our future work is to explore and realize the actuating ability of CFRCLE composites, which is also the original goal of this paper. In this study, when the temperature is not high enough to cause significant deformation of the structure, the modulus of LCE decreases seriously. It leads to that although the structure has achieved bearing capacity and deformation capacity, the bearing capacity at higher temperature is not excellent. There are some methods may improve the actuating ability of composite materials, such as replacing liquid crystal materials with photoinduced deformation LCE. Some methods are expected to improve the actuating ability of the composites, such as replacing liquid crystal materials with photoinduced deformation LCE, or replacing fiber materials with continuous SMP fibers with deformation ability. Once the actuation capability is achieved, this research can be used to develop new avenues for creating soft robotics, mechanical metamaterials, and artificial muscles.”

References

- 1 Guo, Y., Zhang, J., Hu, W., et al. Shape-programmable liquid crystal elastomer structures with arbitrary three-dimensional director fields and geometries. *Nat. Commun.* 12 5936 (2021).
- 2 Peng, X., Wu, S., Sun, X., et al. 4D Printing of Freestanding Liquid Crystal Elastomers via Hybrid Additive Manufacturing. *Adv. Mater.* 34 2204890 (2022).
- 3 Waters, J. T., Li, S., Yao, Y., et al. Twist again: Dynamically and reversibly controllable chirality in liquid crystalline elastomer microposts. *Sci. Adv.* 6 eaay5349 (2020).
- 4 Kotikian, A., McMahan, C., Davidson, E. C., et al. Untethered soft robotic matter with passive control of shape morphing and propulsion. *Sci. Robot.* 4 eaax7044 (2019).
- 5 Zuo, B., Wang, M., Lin, B., et al. Visible and infrared three-wavelength modulated multi-directional actuators. *Nat. Commun.* 10 4539 (2019).
- 6 Han, M. & Ahn, S. Blooming Knit Flowers: Loop-Linked Soft Morphing Structures for Soft Robotics. *Adv. Mater.* 29 1606580 (2017).
- 7 Miao, J., Ge, M., Peng, S., et al. Dynamic Imine Bond-Based Shape Memory Polymers with Permanent Shape Reconfigurability for 4D Printing. *ACS Appl. Mater. Interfaces* 11 40642 (2019).
- 8 Gladman, A. S., Matsumoto, E. A., Nuzzo, R. G., et al. Biomimetic 4D printing. *Nat. Mater.* 15

413 (2016).

9 Guin, T., Settle, M. J., Kowalski, B. A., et al. Layered liquid crystal elastomer actuators. *Nat. Commun.* 9 2531 (2018).

10 Liu, S., Li, Y., and Li, N.. A novel free-hanging 3D printing method for continuous carbon fiber reinforced thermoplastic lattice truss core structures. *Mater Design* 137 235 (2018).

Reviewer #2

Please see my attachment file for your recommendations.

The authors studied the influence of (spatial) 3D printing parameters on liquid crystal elastomer encompassing continuous Fiber. Here they claim that both the printing speed and light source for curing have an outstanding role in spatial 3D printed samples; also, the composite containing off-center Fiber shows better results regarding mechanical features. There is no clear-cut hypothesis or deep discussion in the current version of the paper, especially for chemical analysis and simulation parts. Despite that, this manuscript's novelty and originality are a little above-average, but it needs major modification to be ready for publishing in *Nature Communications*.

Answer: Thank you for the comments on the paper. We have added some experiments and simulations to analyze the mechanism of CFDIW 4D printing. We have revised the manuscript and redrew the figures as suggested. The specific modifications are as follows:

I) The mechanism of interfacial properties and mechanical performances of LCE composites have been analyzed in greater depth, and it has been proved that off-center distribution of fibers has an improvement on interfacial properties and some mechanical performances.

II) SEM observation experiment for impregnation effect and FTIR experiment for verifying reaction mechanism have been supplemented.

III) The simulations of thermal deformation and compression behavior of three-dimensional structures have been supplemented, and the simulation model has been optimized.

IV) Major revisions have been made to the manuscript to supplement the introduction of the CFDIW process mechanism and some technical details.

V) Videos of 4D printing process and deformation process have been added, and photos of deformation effects and fiber off-center effects have been added.

VI) All the figures have been modified.

1) Please be more specific in your title (it is charming now but not clear): I think it is not 4D printing; just replace 3D printing.

For instance,

Understanding Spatial Deformation by Controllable Off-center Continuous Fiber 3D Printing

Answer: Thank you very much for this greatly appreciated suggestion. The title has been revised to be clearer, and some keywords such as “Liquid Crystal Elastomer”, “freestanding” have been added to the title. The new title is:

“Programmable Spatial Deformation by Controllable Off-center Freestanding 4D Printing of Continuous Fiber Reinforced Liquid Crystal Elastomer Composites”

In addition, the process proposed in this manuscript is 4D printing. We are sorry to cause

your confusion because the text and figures in the unrevised manuscript are not clear enough.

In fact, the samples we prepared (including composite filaments and truss structures) have the ability to produce deformation with temperature changes. We have added the video (Movie S3) to show the deformation process. The frames with $t = 0$ s, 4 s and 23 s in the video were shown in Figure 1, illustrating the continuous increase of curvature of the composite in the ambient temperature of 150 °C. This shows that the samples we prepared are 4D printed structures.

Figure 1 The 0 s, 4 s, and 23 s in the video of the deformation process

2) There are some typo mistakes in the manuscript. Please take care of them, like Page 12, lines 251 and 252, ... The molecular orientation has a significant influence on the molecular chain orientation?????...

Answer: We are sorry that there are some typo mistakes in the manuscript. We revised the sentence in lines 251 and 252 as follows:

“The molecular orientation has a significant influence on the thermal shrinkage property.”

In addition, we found other typo mistakes in the manuscript and revised them. For example, some "tension force" in the manuscript has been replaced by "lateral force", because the direct force causing fiber off-center distribution is the lateral force; "eccentricity" in the manuscript has been replaced by "aspect ratio", because the former can only describe sections of elliptical shape, while the latter can describe sections of various shapes.

3) Material selection:

What is the material of Fiber? It is not clear at all.

What is the rationale behind the use of this LCE instead of a well-known matrix?

Answer: Your suggestion is greatly appreciated.

The continuous fiber material we use is aramid fiber (Sovetl, China). We added the following content in *Methods*:

"The continuous fiber material used in this work is aramid fiber (Sovetl, China), which has an axial CTE of $-2 \times 10^{-6} \text{ K}^{-1}$, a tensile strength of 3.6 GPa and a tensile modulus of 131 GPa."

The other question is why we chose LCE in this work. In fact, there are many kinds of 4D printing matrices that we have considered, such as thermoplastic resins, shape memory polymers (SMP), and hydrogels.

Our previous works are related to the 4D printing for composites with continuous fiber reinforced thermoplastic matrix, and the matrix material with better deformation effect is PA-based composite. However, its deformation ability is far inferior to that of LCE matrix

composites. When the temperature is heated to 150 °C, the curvature change of the continuous fibers reinforced PA composite is about 0.028 mm⁻¹, while the curvature change of the continuous fibers reinforced LCE composite used in this paper is about 0.33 mm⁻¹.

We also considered SMPs as matrix materials due to their excellent mechanical properties. However, their deformation process is not continuous. SMPs can only change the status directly from temporary shape to final shape, and the deformation process is usually irreversible. [1, 2]

Hydrogels are also often used as 4D printing materials due to their excellent deformability, and have been studied in our previous exploration. However, our experiments showed that the interface bonding effect between the hydrogel and the continuous fiber is not good because the surface of the hydrogel is very wet.

LCE material has advantages in both thermal deformation ability and mechanical property [3], so it is selected as the matrix for 4D printing in our research, even though the 4D printing for freestanding continuous fibers reinforced liquid crystal materials is a great challenge.

We have added the consideration of material selection into the introduction part of this paper, as follows:

“LCEs are smart materials with high deformation ability, relatively significant mechanical performance, rapid actuation and reversible deformation process that can be prepared by direct ink writing (DIW) 4D printing, and can respond to various external stimuli, such as light, heat, and electric fields.”

“As a material with programmable molecular chain orientation, LCE is expected to produce the freestanding, bearable and deformable three-dimensional structure.”

4) You should mention the mechanism of mechanical strengthening and present a convincing description of "why fiber is able to be wet by LCE?". In the current version, the authors merely reported the numbers and observations. So please evaluate the chemical analysis of Fiber and understand the role of interaction with the matrix.

Answer: Your suggestion is greatly appreciated. According to previous works and our experimental results, we consider that there is no observable chemical reaction between liquid crystal material and aramid fiber in this study. The combination effect of these two materials mainly comes from the liquid crystal fully impregnated on the surface of the fiber.

In the revised manuscript, we supplemented the observation of the SEM images of the cross sections of the composites, which showed that the liquid crystal penetrated into the interior of the fiber bundle, and proved that these two materials have good interface characteristics.

Figure 2 SEM images of the cross section of the composite with good impregnation effect

We have added a description of how to improve the wetting effect through fiber off-center distribution in the manuscript, as follows:

“In the process of composite forming, the off-center distribution is helpful for the liquid crystal to impregnate on the fiber surface. When the lateral force of the nozzle is low during the pulling process, the cross section of the fiber is close to the circle, and there are some cavities not filled with liquid crystal inside (Fig. S5). The higher off-center effect makes the cross-section shape of the fibers more flat, so that the liquid crystal is easier to penetrate into the interior of the fiber bundle. The cross section of the pulled composite with a high off-center effect was observed by SEM, as shown in Fig. 1e, indicating that the impregnation of the two materials was sufficient.”

4) As clear as crystal, the continued Fiber can improve the mechanical properties. However, you just study static mechanical properties regardless of dynamic properties: Is it true?. Please imply dynamic phenomenon with the aid of conducting DMTA test in the manuscript and provide reasonable reasons for the mechanical improvement and chain orientation. For instance, the polymer chain moves more easily in XX sample because of the printing angle and free volume and slippage on the surface of Fiber that is able to enhance the chance of fiber pull-out.

Answer: Your suggestion is greatly appreciated. In this work, we focus on the static properties of composite materials and truss structures rather than the dynamic properties, because the materials prepared by our 4D printing process are difficult to prepare into shapes that meet the dynamic analysis.

Take the DMA test as an example, the material required is usually sheet-like, such as a 30mm * 7mm * 0.5 mm cuboid structure, but the structure we prepared is a cylindrical wire with a diameter of 1 mm to 2 mm, which cannot meet the requirement of DMA test.

5) Is it either a 3D printed sample or a 4D printed (read about them)?

One of the most effective parameters in your sample is "Programmable Spatial Deformation", which is overlooked in this study. What is the meaning of programmable? I could not find any 4D printed sample that has the capability of changing its shape upon stimuli field. Deformation of samples due to curing time is not controllable and acceptable as a 4D printing.

Answer: We are sorry to cause your confusion because the text and figures in the unrevised manuscript are not clear enough.

In fact, the samples we prepared (including composite filaments and truss structures) have the ability to produce deformation with temperature changes. We have added the video (Movie S3) to show the deformation process. The frames with $t = 0$ s, 4 s and 23 s in the video were shown in Figure 1, illustrating the continuous increase of curvature of the composite in the ambient temperature of 150 °C. This shows that the samples we prepared are 4D printed structures.

The other question is whether the prepared structure reflects "programmable". Our answer is "Yes". Specifically, programmability is reflected in the fact that we can simultaneously control the deformation curvature and deformation direction of the structure during the 4D printing process. In our revised manuscript, the second subsection (Controlling the off-center degree of the fibers) of the results section introduces the method of adjusting the deformation amplitude by the

off-center degree, and the third subsection (Controlling the off-center directions of the fibers) of the results section introduces the method of adjusting the deformation direction by the off-center position.

6) In the majority of figures and results, simulation is spotlighted. I suppose you can find more details of the simulation part in ABACUS; see other papers.

Answer: Your suggestion is greatly appreciated. We have added details of simulation analysis in the Methods section:

“Simulation. The finite element analysis of the thermo-induced deformation process of continuous fibers reinforced LCE composites was performed by ABAQUS. The simulation included the parameters: the elastic modulus, the Poisson’s ratio and the CTE. The LCE structure in the simulation were assumed to have great thermal shrinkage property, and the initial temperature was set to room temperature (25 °C). The composite truss was modeled as a cylinder, and the cross-section of reinforced continuous fiber bundle was modeled as a circle and an off-center bow, which were used to simulate the case of low and high off-center degree, respectively. The specific shape parameters were obtained from the SEM images shown in Figure 3. Second-order C3D8I elements were used during the simulations. In the analysis of the pyramid-shaped structure, initial geometric imperfections were introduced to the mesh to better trigger the torsional deformation behavior of the structure.”

In addition, we optimized the simulation analysis and adopted a model with higher accuracy. In the previous simulation analysis, cylindrical liquid crystal material and elliptical continuous fiber material were used to model a single wire, while the model of truss structure was approximate. In the simulation analysis of the revised manuscript, the elliptical cylinder model with higher accuracy was adopted for all the structures, and the thermal deformation simulation analysis of the pyramid-shaped structure has been added.

7) Due to lacking controversial debits in the curing part, I would postpone the in-depth science-oriented discussion to the revision stage. Only one-point springs to my mind: you tried to cure it and connect all the macromolecules. Please make sure that outstanding mechanical behavior comes from which source: curing or Fiber.

Answer: Thank you very much for this greatly appreciated suggestion. Although the curing degree of the LCE and the existence of continuous fiber both provide the mechanical properties of the composite structure, continuous fiber is the main factor to enhance the mechanical properties.

In order to analyze the improvement of mechanical properties of fiber materials, we tested the tensile properties of liquid crystal materials without continuous fibers. When the light curing time is 10 min, the average tensile modulus of pure LCE material is 0.24 GPa. In comparison, the tensile modulus of continuous fiber reinforced LCE is 2.8 GPa, with an increase of about 10.7 times.

In the revised manuscript, we introduced the improvement of fiber on the mechanical properties of LCE, as follows:

"In comparison, the tensile modulus of LCE material without continuous fibers is only 0.24 GPa when the light curing time is 10 min, indicating the improvement of the mechanical property by continuous fibers."

References

- 1 Han, M. & Ahn, S. Blooming Knit Flowers: Loop-Linked Soft Morphing Structures for Soft Robotics. *Adv. Mater.* 29 1606580 (2017).
- 2 Miao, J., Ge, M., Peng, S., et al. Dynamic Imine Bond-Based Shape Memory Polymers with Permanent Shape Reconfigurability for 4D Printing. *ACS Appl. Mater. Interfaces* 11 40642 (2019).
- 3 Gladman, A. S., Matsumoto, E. A., Nuzzo, R. G., et al. Biomimetic 4D printing. *Nat. Mater.* 15 413 (2016).

Reviewer #3

This manuscript presents a new approach to printing temperature-responsive LCE composites consisting of a soft LCE matrix as a shell and rigid aramid fiber bundles as a core. Intriguingly, the authors developed a customized direct-ink-writing apparatus that can simultaneously print the aramid fiber cores and LCE shell. During the printing process, the aramid fibers were intentionally distributed off-center within the composite fibers to induce bending deformation based on the mismatch of CTE values between the LCE matrix and aramid fiber. The author claimed that the location of aramid fibers can be precisely controlled to programmable bending. While the concept of off-center continuous LCE composite fibers is quite unique and interesting, the current manuscript is not technically sound because the fundamental deformation mechanism of the fiber as well as the printing process is not clear. Also, the quality of the manuscript including figures/figure captions and details of explanation is not satisfactory. Therefore, I recommend the authors should refine the manuscript, and submit it elsewhere. The following comments could improve the quality of this manuscript.

Answer: Thank you for the comments on the paper. We have added some experiments and simulations to analyze the mechanism of CFDIW 4D printing. We have revised the manuscript and redrew the figures as suggested. The specific modifications are as follows:

I) The mechanism of interfacial properties and mechanical performances of LCE composites have been analyzed in greater depth, and it has been proved that off-center distribution of fibers has an improvement on interfacial properties and some mechanical performances.

II) SEM observation experiment for impregnation effect and FTIR experiment for verifying reaction mechanism have been supplemented.

III) The simulations of thermal deformation and compression behavior of three-dimensional structures have been supplemented, and the simulation model has been optimized.

IV) Major revisions have been made to the manuscript to supplement the introduction of the CFDIW process mechanism and some technical details.

V) Videos of 4D printing process and deformation process have been added, and photos of deformation effects and fiber off-center effects have been added.

VI) All the figures have been modified.

In particular, we would like to thank you for your suggestions on the evaluation of the

impregnation effect. Therefore, we found that the impregnation effect of the composite can be changed by adjusting the off-center degree of the fiber bundle, and the impregnation effect can be improved by increasing the off-center degree. In the unrevised manuscript, we only focus on the influence of off-center fiber distribution on deformation performance and bearing capacity.

1) Page 4: the authors claim that one of the challenges of 4D printed structure is poor mechanical bearing capacity caused by the material and shape restrictions. However, some of the examples listed in page 4 is not so related mechanical bearing capacity of 4D printed materials. The logic of this paragraph should be revisited.

Answer: Your suggestion is greatly appreciated. In fact, it is precisely because the existing LCE 4D printing is difficult to achieve good bearing performance, so relevant research usually only focuses on the actuating characteristics. In this section, we analyzed that the existing 4D printing structure cannot achieve the bearing capacity because most LCE structures cannot achieve freestanding. Although Peng et al. [1] achieved freestanding 4D printing, the LCE prepared was made of filamentous material, and could not carry.

2) Can the authors provide any results of the interfacial adhesion between LCE and aramid fiber?

Answer: Your suggestion is greatly appreciated. In order to verify the impregnation effect of the composites, we observed the SEM images of the composite cross section with higher magnification, as shown in the following figure. These new SEM results were added to the article and *Supplement Information*.

Figure 1 SEM images of the cross section of the composite with good impregnation effect

Figure 2 SEM images of the cross section of the composite with poor impregnation effect

As shown in Figure 2, when the off-center degree of the fiber is low during the 4D printing process, the cross section of the fiber bundle is approximately circular, and the liquid crystal is difficult to immerse into the interior of the fiber bundle. The structures with different printing speeds were observed by SEM, and the impregnation effect could not be improved. However, when we reduce the inclination angle in the printing process to -10° , the impregnation effect of the material is significantly improved. The SEM image of the cross section of the prepared composite structure is shown in Figure 1. It can be seen that the cross section of the fiber bundle becomes flat, which makes it easier for the liquid crystal to immerse into the interior of the fiber bundle, and no obvious holes can be observed in the SEM image. In summary, the impregnation effect can be controlled by adjusting the degree of fiber off-center degree.

We have added a description of how to improve the wetting effect through fiber off-center distribution in the manuscript, as follows:

“In the process of composite forming, the off-center distribution is helpful for the liquid crystal to impregnate on the fiber surface. When the lateral force of the nozzle is low during the pulling process, the cross section of the fiber is close to the circle, and there are some cavities not filled with liquid crystal inside (Fig. S5). The higher off-center effect makes the cross-section shape of the fibers more flat, so that the liquid crystal is easier to penetrate into the interior of the fiber bundle. The cross section of the pulled composite with a high off-center effect was observed by SEM, as shown in Fig. 1e, indicating that the impregnation of the two materials was sufficient.”

3) There is no information on aramid fiber such as chemical structure, molecular weight, vendors, etc.

Answer: Your suggestion is greatly appreciated. We have added the information of aramid fiber in *Methods* and *Supplement Information*, including chemical structure, mechanical properties, coefficient of thermal expansion, vendor, etc.

We added the following content about the aramid fiber we used in *Methods*:

"The continuous fiber material used in this work is aramid fiber (Sovetl, China), which has an axial CTE of $-2 \times 10^{-6} \text{ K}^{-1}$, a tensile strength of 3.6 GPa and a tensile modulus of 131 GPa."

4) It would be much helpful if the authors can provide videos that describe the printing process and thermal actuation of each printed structure.

Answer: Your suggestion is greatly appreciated. We have added videos of the printing process of composite materials, as shown in Movie S1, and S2.

In addition, we have added the video (Movie S3) to show the deformation process. The frames with $t = 0 \text{ s}$, 4 s and 23 s in the video were shown in Figure 3, illustrating the continuous increase of curvature of the composite in the ambient temperature of $150 \text{ }^\circ\text{C}$. This shows that the sample have good deformation ability.

Figure 3 The 0 s, 4 s, and 23 s in the video of the deformation process

5) The simulation result of Fig 2d does not show much detail and the importance of the result. Also, there is no information regarding the color distribution in Fig 2d.

Answer: Thank you very much for this greatly appreciated suggestion. We are sorry that we didn't clearly express our intention to use this simulation image. In fact, we use this figure to show the deformation shape of the structure. Different colors represent different degrees of deformation displacement, red represents higher displacement values, and blue represents lower displacement values. We have added this explanation in the manuscript.

We modified this figure from the following aspects: i) the shape before deformation was added and compared with the shape after deformation, indicating the deformation process of the structure; ii) the *Simulation* subsection is added to the methods section of the manuscript to introduce the simulation information.

Figure 4

6) What is the “m” in Fig 1a? Please provide details because the authors seem to use a mixture of R6M and RM257. What is the molar ratio of the LC monomer? Also, what is the molar ratio between the LC monomers and the thiol-chain extender? – this is a critical molecular parameter to determine materials' properties. Also, the diacrylate-terminated LC oligomer structure in Fig 1a is not correct – please check out the reaction details for this chemistry reported by T. White (Macromolecules 2021, 54, 23, 11074–11082)

Answer: The letter *m* represents the number of carbon-carbon single bonds inside brackets. For R6M, $m=6$; and for RM257, $m=3$. In our experiments, the molar ratio of R6M to RM257 is 1.0:0.3, and we have supplemented this information in the manuscript. The redrawn figure of the molecular structural formulas is shown below.

Figure 5 Molecular structures

We have studied the article you recommended to us and gained a deeper understanding of the reaction mechanism, and the redrawn figure of the reaction mechanism is shown in *Supplementary Information*:

Figure 6 The reaction mechanism

7) Please redraw of Fig 1c with x-axis with log-scale. Many 3D or 4D printed LCE papers used the log scale for both x- and y-axes.

Answer: Your suggestion is greatly appreciated. When testing shear viscosity, the sampling data interval is about $1\text{-}2\text{ s}^{-1}$. Therefore, if both the x-axis (shear rate) and the y-axis (viscosity) are modified to be log-scale, the left half of the curve will be very flat, while the right half will fluctuate severely, as shown in the following figure:

Figure 7 Both the x-axis and the y-axis are modified to be log-scale

Therefore, we redrew the figure and the y-axis were modified to be log-scale. The new figure is shown as bellow:

Figure 8 Only the y-axis is modified to be log-scale

8) The theta, alpha and beta are not presented in Fig 3. Please include this information in Fig 3.

Answer: Your suggestion is greatly appreciated. We have redrew Figure 3, which represents the information of theta, alpha and beta. The new figure is shown bellow:

Figure 9 Annotations for alpha and beta

9) SEM images in Fig 3g do not clearly support off-center positioned aramid fiber. The resolution should be much improved, and higher magnification images should be provided as well.

Answer: Thank you very much for this greatly appreciated suggestion. The control effect of fiber off-center distribution cannot be fully demonstrated by SEM images alone, because the material used for SEM observation is a small segment cut from the composite wire. Therefore, we took a clearer picture of composite wire to prove that the off-center position of the fiber can be adjusted.

Because the color of aramid fiber and liquid crystal are relatively close, it is not convenient to view, so the continuous fibers in the photos in the revised manuscript are enlarged and marked. The photo is shown below:

Figure 10 The reshooted photo

10) The raw data for 2D-XRD images should be provided, especially azimuthal plots.

Answer: Your suggestion is greatly appreciated. We directly generated 2D patterns through XRD testing, and used Fit2D software to calculate the orientation degrees of the samples. We have provided the complete 2D patterns in *Supplement Information*. In the manuscript, we present 2D-pattern results in order to make it easier for readers to compare the orientation degrees of liquid crystal materials under different 4D printing parameters.

11) The experimental conditions for XRD and SEM are not provided.

Answer: Your suggestion is greatly appreciated. The experimental conditions of XRD and SEM are included in the methods section of our manuscript. Besides, we have expanded these information of SEM as follows:

“**SEM.** The composite filament was cut into sections, and the cross section is used to observe the off-center distribution and the impregnation effect of the fibers. The SEM images were taken by a electron microscope (Hitachi, SU3500, Japan) using a voltage of 5 kV. Before observation, the samples were sputtered with gold in vacuum for 120 seconds with a current of 40 mA to enhance the conductivity of their surfaces.”

“**2D XRD.** To obtain the molecular chain orientation of liquid crystal components in composite trusses, the continuous fibers were stripped from the composites and ten 30-mm long LCE filaments were cut and bundled for XRD test (Rigaku, HomeLab, Japan). The anode target X-ray source used was a Cu K α radiation with a maximum output power of 2.97 kW and an electron beam focal spot diameter of 70 μ m, and the detector used was a Hypix-6000 photon direct reading detector. The Hermans orientation parameter was calculated using the azimuthal integration from Fit2D software, and the following equation is used to calculate the orientation degree of the polymer:

$$P = 1 - \frac{3 \int_0^{2\pi} I(\theta) [\sin^2 \theta + \sin \theta \cos^2 \theta \ln(1 + \tan \theta)] d\theta}{2 \int_0^{2\pi} I(\theta) d\theta} \quad (3)$$

where $I(\theta)$ represents the signal intensity in different directions on the XRD pattern.”

12) What is the reason for using a fluorescent agent?

Answer: Your suggestion is greatly appreciated. In fact, the function of fluorescent agent is only to enable us to take clearer pictures of composite materials. Because the content of fluorescent agent is very low, it hardly affects the deformation ability and mechanical properties of the structure. We refer to some previous works on the types and proportions of fluorescent agents [2].

Because the fluorescent agent is not the main material that determines the performance of LCE, in the revised manuscript, we do not mention fluorescent agent in the *Results* section, but introduce the information of fluorescent agent in the *Methods* section.

13) In Fig 2, the authors show the schematic for the printing apparatus. But I think more details should be provided for each part. The printing procedure presented in the manuscript and Fig 2 is not enough to completely understand the exact principle of printing process and the way to control the location of the aramid fiber core.

Answer: Thank you very much for this greatly appreciated suggestion. In the revised manuscript, we revised the schematic diagram to describe the mechanism of controlling the location of the aramid fiber core, as shown in Figure 11, and we have expanded and revised the description of methods for controlling the fiber position:

Figure 11 The method for controlling the off-center position of the fibers

The revised description in the manuscript is as follows:

“The advantage of the CFDIW 4D printing process is that it can control the formation structure of the truss structure and the distribution of the fibers in the truss simultaneously. If no additional control measures are adopted, the fibers will be distributed on the upper surface of the composite under the action of tension. However, the deformation of three-dimensional structure requires that the fibers are distributed at any position within the composite filament cross section, which can be achieved by adjusting the moving path of the nozzle. A composite truss with the fibers distributed on the left side was used as an example to demonstrate the control method of the off-center position. To form an adjustable off-center composite filament, the movement path of the nozzle includes pulling out, lifting and falling motions, as shown in Fig. 4a. The pulling out step was used to determine the fiber off-center direction while “spitting out” the composites, and the falling movement was used to determine the forming position of the truss. Specifically, the composite was pulled out of the nozzle with the fibers distributed at the top of the filament, and the filament was cured using an area light source except for its two ends. Then the nozzle was lifted along an arc trajectory, causing the filament to rotate in the x-z plane until it was vertically upwards, so that the fibers were distributed on the left side. Due to the two ends of the filament had not yet solidified, the rotational motion of the filament was not constrained. After the nozzle had fallen along an arc trajectory in the y-z plane, the end point of the fiber was finally cured using a higher-power point light source, and the off-center angle of the fibers had been adjusted (Fig.4c).”

14) Fundamental deformation mechanism of the composite fiber should be more clearly presented. I do not think this was not thoroughly explained in the paragraph or in the figure. This is important because this will eventually explain the deformation process of other complex printed structures.

Answer: Thank you very much for this greatly appreciated suggestion. We have reintroduced the deformation mechanism from the following aspects:

I) We readjusted the structure of the *Results* section. The mechanism of bending deformation of the composite is introduced in the first subsection; the control mechanism of deformation degree is introduced in the second subsection; the control mechanism of deformation direction is

introduced in the third subsection.

II) In the first subsection (CFDIW 4D printing method for off-center CFRLCEs), we introduce the difference of CTEs between continuous fiber and LCE and the off-center distribution of fiber as the cause of deformation, and add the schematic diagram of the method for producing off-center distribution.

III) In the second subsection (Controlling the off-center degree of the fibers), we introduce that the off-center degree of fiber is the main factor affecting the deformation degree of composite materials. The influence of off-center distribution is reflected in two aspects, one is the change of the geometry shape of the fiber bundle, and the other is better impregnation effect.

IV) In the third subsection (Controlling the off-center direction of the fibers), we introduce that the off-center direction of fiber is the main factor affecting the deformation direction of composite materials. We redrew the schematic diagram to describe the mechanism of controlling the off-center direction.

References

- 1 Peng, X., Wu, S., Sun, X., et al. 4D Printing of Freestanding Liquid Crystal Elastomers via Hybrid Additive Manufacturing. *Adv. Mater.* 34 2204890 (2022).
- 2 Kim, Y., Yuk, H., Zhao, R., et al. Printing ferromagnetic domains for untethered fast-transforming soft materials. *Nature* 558 274 (2018).

Reviewer #4

Some recent papers have reported the 4D printed liquid crystal elastomer using the direct ink writing method. However, this work introduced continuous fibers in the liquid crystal elastomer using a direct ink writing method for 4D printed spatial structures with good mechanical properties. Although the paper contains interesting work, some major questions need to be answered. Here are specific points for consideration:

Answer: Thank you for the comments on the paper. We have added some experiments and simulations to analyze the mechanism of CFDIW 4D printing. We have revised the manuscript and redrew the figures as suggested. The specific modifications are as follows:

I) The mechanism of interfacial properties and mechanical performances of LCE composites have been analyzed in greater depth, and it has been proved that off-center distribution of fibers has an improvement on interfacial properties and some mechanical performances.

II) SEM observation experiment for impregnation effect and FTIR experiment for verifying reaction mechanism have been supplemented.

III) The simulations of thermal deformation and compression behavior of three-dimensional structures have been supplemented, and the simulation model has been optimized.

IV) Major revisions have been made to the manuscript to supplement the introduction of the CFDIW process mechanism and some technical details.

V) Videos of 4D printing process and deformation process have been added, and photos of deformation effects and fiber off-center effects have been added.

VI) All the figures have been modified.

1. Why are the fibers beneficial to support the direct manufacture of spatial 3D structures? What is the mechanism? What is the difference with the recent publications on self-Supplement, for example, DOI: 10.1002/adma.202204890?

Answer: Thank you very much for this greatly appreciated suggestion. Like most resins, the moduli of LCEs are much lower than those of continuous fibers, so they are relatively soft. The tensile moduli of the continuous fibers are very high, but monofilaments inside the fiber bundles are not bonded, so the fibers are not freestanding. Fiber/resin composites have high moduli, and the internal fibers can be bonded together by impregnated resin, so the freestanding composites can be realized. In fact, 3D printing of freestanding continuous fibers reinforced thermoplastic resin has been reported [1].

The other question is the difference with the recent publication on self-supporting.

The paper (DOI: 10.1002/adma.202204890) entitled *4D Printing of Freestanding Liquid Crystal Elastomers via Hybrid Additive Manufacturing* [2] was published on *Advanced Materials* in August 2022, which realized the freestanding of liquid crystal materials. It uses multiple high-power lasers to aim at the exit direction of printing, which greatly reduces the curing time of liquid crystal. Peng et al. [2] discusses why it is difficult for LCE to prepare three-dimensional structure as follows:

“Although 3D structures, such as pinecone and saddle-shaped structures, can be achieved by 2D structures via different actuation strains between layers, the layer-by-layer manner of material deposition in DIW makes LCEs to be printed on the build platform or the previous layers. As a result, the actuation of the printed LCE structures is limited to planar shrinkage, simple bending, or twisting.”

“Still, this approach only aligns mesogens in one direction; hence, applications are limited to thin films and simple bending actuation.”

Our manuscript was submitted in October 2022, less than two months later than the publication of this paper, and we did not notice and cite this work at that time. Now the revised version have cited this study and made full comparison with our work.

Although their work [2] and our work have both realized the 4D printing for freestanding LCE, there are still many differences between us: i) In their work, additional support structures need to be provided to fix both ends of the liquid crystal filaments, and different 4D printed structures need to rely on different support structures, while our 4D printing process does not need support structures; ii) Continuous fiber is introduced into our 4D printing materials, which makes the structure have better bearing capacity; iii) Our work does not require expensive multiple laser sources, thus reducing the printing cost.

In our revised manuscript, the progress of 4D printing of the three-dimensional LCE structure is re-introduced as follows:

“Peng et al. realized the 4D printing of the freestanding LCE structure by using multiple laser sources to cure the LCE in-situ when the nozzle moves and extrudes the liquid crystal material. However, this method has the following limitations: i) the forming process of LCE structures depends on Supplement structures; ii) The LCE truss structure has little bearing capacity; iii) The solidification of liquid crystal materials requires expensive high-power multiple laser sources.”

2. The mechanical properties of the polymer matrix can be certainly improved with the continuous fibers, such as the traditional carbon fiber reinforced polymer matrix laminates and the 3D printed continuous fibers reinforced polymer-matrix composites. This work also utilized continuous fibers to improve the mechanical properties of the liquid crystal elastomer, however, the off-center distribution of continuous fibers is harmful to the mechanical properties, how to avoid it and make the on-center distribution?

Answer: Thank you very much for this greatly appreciated suggestion. In the CFDIW process, in order to make the distribution of continuous fibers in the composite on-center, the printer should be redesigned. The piston should be changed to be coaxial with the nozzle, and the inclination angle of the fiber during the pulling process is 90° .

Even so, off-center continuous fiber does not mean damage to mechanical properties. We used ABAQUS to make a set of filament models for verification.

For the pyramid model in Figure 8c, we additionally analyzed a on-center distributed composite structure for comparison, and the composite materials with different distributions of the fiber bundle showed different bearing effects. It can be seen that the bearing effect of the concentrically distributed structure is not as good as that of the structure with the fibers off-center distributed on the upper surface.

Fibers off-center distributed on the side

Fibers off-center distributed on the upper surface

Fibers on-center distributed

We have added the following content to the manuscript to illustrate the mechanical improvement effect of the off-center distribution of fibers:

“The CFDIW process enables 4D printing for free-standing CFRLCE trusses, making it possible to increase the bearing capacity of composite structures. Due to the CFDIW process is not used to prepare composites with on-center distributed fibers, it is necessary to analyze the impact of fiber distribution on mechanical properties through simulation. When maintaining other printing parameters such as fiber content unchanged, the distribution of fibers hardly has a significant impact on the composite filaments. Therefore, mechanical analysis of the bending behavior of the structure is emphasized. Here, a pyramid structure as shown in Figure 8a is used for ABAQUS simulation analysis. The fibers inside the four filaments at the top of the pyramid structure are distributed on-center and off-center (with eccentric angles of 0 or 90 degree), and a vertical downward force is applied to trusses at the top. When the applied displacement is one fifth of the height of the structure, the shapes of the structures are shown in Figure 8b. It can be clearly seen that when fibers off-center distributed at the top, the truss produces almost no torsional behavior, while when the fibers off-center distributed at the side or on-center, the truss undergoes severe torsion, which leads to earlier instability of the structures. The reason for the difference in mechanical behavior is that when the fiber off-center direction is at the top, the larger aspect ratio of the fiber cross-section contributes to a better resistance to horizontal torque, and thus can effectively avoid structural failure due to torsion, while on-center or laterally off-center distributed fibers cannot resist this torsion. This theoretically proves that the off-center distribution of fibers can improve mechanical properties of the CFRLCE structures.”

3. The potential applications and prospects of 4D printed continuous fiber-reinforced liquid crystal elastomer composites are unclear. Please clarify them.

Answer: Your suggestion is greatly appreciated. The potential applications of this technology are artificial muscle and actuator with better performance, which is also the original goal of this paper. We have improved the deformability and mechanical properties by introducing continuous fibers to achieve this goal. When the 30 mm long straight composite truss is heated to 150 °C, the minimum equivalent shrinkage ratio of 20% can be reached through bending deformation, and the axial elastic modulus of the fiber composite is increased by an order of magnitude (from 0.24 GPa to 2.8 GPa). However, the modulus of the liquid crystal elastomer decreases significantly when it is heated to 150 °C. Therefore, although the truss involved in this study has achieved the bearing

and deformation capacity, it has not achieved the actuating capacity. However, it can be predicted that if the liquid crystal material is replaced with other response modes, such as photoinduced deformation, the composite will be expected to achieve a higher level of actuation capability.

The discussion of potential applications was added to the discussion part of the manuscript:

“Our future work is to explore and realize the actuating ability of CFRCLE composites, which is also the original goal of this paper. In this study, when the temperature is not high enough to cause significant deformation of the structure, the modulus of LCE decreases seriously. It leads to that although the structure has achieved bearing capacity and deformation capacity, the bearing capacity at higher temperature is not excellent. There are some methods may improve the actuating ability of composite materials, such as replacing liquid crystal materials with photoinduced deformation LCE. Some methods are expected to improve the actuating ability of the composites, such as replacing liquid crystal materials with photoinduced deformation LCE, or replacing fiber materials with continuous SMP fibers with deformation ability. Once the actuation capability is achieved, this research can be used to develop new avenues for creating soft robotics, mechanical metamaterials, and artificial muscles.”

4. In Fig. 7, the 4D printed truss in this work could withstand up to 2805 times its own weight. It is not a surprising result, and 4D printed truss with many continuous fibers reinforced polymers can also realize a similar result. Besides, the bearing capacity is not the important research point for 4D printed liquid crystal elastomer (active material), and the authors should give a surprising actuation capability result of 4D printed liquid crystal elastomer instead of the bearing capacity.

Answer: Your suggestion is greatly appreciated. We noticed that in previous works, the actuating capacity of LCEs often cannot match their bearing capacity [2, 3]. Therefore, we tried to explore an LCE composite with wider applicability.

Our original intention is to prepare liquid crystal composites with bearing and actuating functions. Although the liquid crystal can not achieve good actuating performance due to its softening at high temperature, we have also made some new achievements, such as preparing composite materials with better deformation performance and bearing capacity, providing a freestanding liquid crystal 4D printing scheme without auxiliary structure, etc. We consider that this work provides an inspiration for 4D printing of liquid crystal materials, and the subsequent work is to develop photo-deformed liquid crystal materials as the matrix of composite materials, so as to finally realize liquid crystal composites with good bearing capacity and actuating capacity.

5. The model-to-part fidelity of 4D printed continuous fiber reinforced liquid crystal elastomer (Fig. 6-7) in this work seems not good.

Answer: Your suggestion is greatly appreciated. Figure 6 and 7 have been rearranged, and some models have also been redrawn. In the new model diagram, we replace the original two-dimensional model with three-dimensional model, and the new model diagram is easier to understand.

At the end of this response, we have attached a revised version of all the figures.

6. The organization of figures in the manuscript is not good.

Answer: Your suggestion is greatly appreciated. We have modified all the figures, as follows:

Figure 1

Figure 2

Figure 3

Figure 4

Figure 5

Figure 6

Figure 7

Figure 8

In Figure 1, the schematic diagram of off-center mechanism is added, and the SEM images of the cross-sections of composites are added to characterize the off-center distribution effect and the impregnation effect.

In Figure 2, the simulation figure is modified and the photos of deformation process are added.

Figure 3 is rearranged.

In Figure 4, the schematic diagram has been modified to show the control mechanism of off-center distribution more clearly.

Figure 5 is rearranged and the XRD images are corresponding to the curves in the graph. In Figure 6, the model diagrams have been modified into more understandable three-dimensional models.

Figure 7 is added in the revised manuscript, illustrating the 4D printing process and deformation effect of freestanding CFLCD trusses.

Figure 8 is rearranged.

References

- 1 Liu, S., Li, Y., and Li, N.. A novel free-hanging 3D printing method for continuous carbon fiber reinforced thermoplastic lattice truss core structures. *Mater Design* 137 235 (2018).
- 2 Peng, X., Wu, S., Sun, X., et al. 4D Printing of Freestanding Liquid Crystal Elastomers via Hybrid Additive Manufacturing. *Adv. Mater.* 34 2204890 (2022).
- 3 Yuan, C., Roach, D. J., Dunn, C. K., et al. 3D printed reversible shape changing soft actuators assisted by liquid crystal elastomers. *Soft Matter* 13 5558 (2017).

REVIEWER COMMENTS

Reviewer #1 (Remarks to the Author):

The authors have been addressed all comments. I recommend accepting the paper in current form.

Reviewer #2 (Remarks to the Author):

The revision has been done carefully. This paper can publish after minor edits.

Just upgrade the title of supporting information.

Also, harmonize the color of the sample in one figure. For example, see the last figure; (a) and (b) are not adjusted.

In addition, if you could introduce some applications in the conclusion part, it would make great progress in this paper.

Reviewer #3 (Remarks to the Author):

The manuscript has been considerably improved in explaining details of the printing and deformation process including supporting videos. This manuscript may be accepted after addressing the following concerns.

1) Can authors quantify the interfacial adhesion between the aramid fiber and LCE by measuring a single fiber pull-out test or micro-bonding test for example? Good and poor impregnation effects have been explained, but no direct support for interfacial adhesion was provided.

2) Azimuthal plots for the XRD results were still not provided.

3) The function of a fluorescent agent should be provided in the main manuscript as well.

4) There are several concerns and mistakes in References in the introduction.

- page 2, line 40: "relatively significant mechanical-performance" is too vague expression. Please be specific. Ref 9 mainly focuses on damping performance of LCE.

- Ref 11 is not related with 4D printing.

- Page 2, line 40: "rapid actuation" is also not appropriate with ref 10 and 11.

- Ref 12 is not related with light-responsive actuation of LCE, but light-induced shape reconfiguration.

- K. Kim et al, Small, 2021, 17, 2100910 should be also provided in Ref 13. This is the first humidity-responsive LCE with 4D printing.

- Ref 17 and 18 are not highlighting the orientation of LCE chains on the mechanical properties.

- Ref 35 is not LCE paper. Please fix it.

- Ref 41 is not relevant with 4D printing.

- Page 4, line 87: "Guo et al," should be "Liu et al"

- Page 5, line 95: there are several LCE papers discussing about structural stability to the compression forces. (1) Advanced Materials, 2022, 34, 2200272. (2) Advanced Healthcare Materials, 2020, 9, 1901136. (3) Advanced Materials, 2020, 32, 2000797. (4) ACS Appl. Mater. Interfaces 2021, 13, 12698.

Reviewer #4 (Remarks to the Author):

Although the previous concerns have been well addressed, some minor questions should be answered. Here are specific points for consideration.

1. In Fig. 4f, the diameter of the composite filaments with various fiber off-center positions is not

uniform, how to control the diameter?

2. The three structures in Fig. 6a, 6e, and 6i are axisymmetric, while the deformation in the simulation results (Fig. 6d, 6h, and 6l) is not axisymmetric. Are the structures deformed in three-dimensional space not two-dimensional plane?

Response to Reviewers

The following is a point-to-point response to the reviewers' comments.

Reviewer #1

The authors have been addressed all comments. I recommend accepting the paper in current form.

Answer: Thank you for your suggestions on this manuscript.

Reviewer #2

The revision has been done carefully. This paper can publish after minor edits. Just upgrade the title of supporting information.

Answer: Your suggestion is greatly appreciated. The title of *Supplementary Information* has been upgraded.

Also, harmonize the color of the sample in one figure. For example, see the last figure; (a) and (b) are not adjusted.

Answer: Your suggestion is greatly appreciated. Some disharmonious figures (such as Figure 8a) have been modified.

In addition, if you could introduce some applications in the conclusion part, it would make great progress in this paper.

Answer: Your suggestion is greatly appreciated. The discussion of applications was added to the discussion part of the manuscript:

“Due to its excellent deformation ability and mechanical properties, the structures prepared by the CFDIW process are expected to be applied in the field of soft robotics with gripping functions, such as soft robotic arms or drug delivery robots.”

Reviewer #3

The manuscript has been considerably improved in explaining details of the printing and deformation process including supporting videos. This manuscript may be accepted after addressing the following concerns.

1) Can authors quantify the interfacial adhesion between the aramid fiber and LCE by measuring a single fiber pull-out test or micro-bonding test for example? Good and poor impregnation effects have been explained, but no direct support for interfacial adhesion was provided.

Answer: Thank you very much for this greatly appreciated suggestion. In order to provide the direct support for interfacial adhesion of the fibers and the liquid crystals, the micro-droplet debonding tests was carried out, and the following content was added to the manuscript:

“The interfacial properties were characterized by utilizing the micro-droplet debonding test, and the CFRLCE composites exhibit a maximum increased interfacial shear strength (IFSS) reaching up to 5.58 MPa.”

“**Micro-droplet debonding test.** The single aramid fiber was tightened and fixed on the sample holder, and micro-droplets were prepared by dipping the fiber in the melted liquid crystals. Micro-droplets were stuck on the crosshead of the interfacial evaluation equipment (Model HM410, Japan), and were debonded from the single fiber by moving the sample holder with the speed of $0.12 \text{ mm}\cdot\text{s}^{-1}$.”

2) Azimuthal plots for the XRD results were still not provided.

Answer: Thank you very much for this greatly appreciated suggestion. The XRD tests of LCEs with different printing speeds and tensile inclination angles were carried out, and the azimuthal plots for the XRD results of the liquid crystals is shown in *Supplementary Information*.

3) The function of a fluorescent agent should be provided in the main manuscript as well.

Answer: Your suggestion is greatly appreciated. The following content was added to the manuscript:

“For imaging purposes, fluorescent agents were added to the composition.”

4) There are several concerns and mistakes in References in the introduction.

- page 2, line 40: “relatively significant mechanical-performance” is too vague expression. Please be specific. Ref 9 mainly focuses on damping performance of LCE.
- Ref 11 is not related with 4D printing.
- Page 2, line 40: “rapid actuation” is also not appropriate with ref 10 and 11.
- Ref 12 is not related with light-responsive actuation of LCE, but light-induced shape reconfiguration.
- K. Kim et al, *Small*, 2021, 17, 2100910 should be also provided in Ref 13. This is the first humidity-responsive LCE with 4D printing.
- Ref 17 and 18 are not highlighting the orientation of LCE chains on the mechanical properties.
- Ref 35 is not LCE paper. Please fix it.
- Ref 41 is not relevant with 4D printing.
- Page 4, line 87: “Guo et al,” should be “Liu et al”

- Page 5, line 95: there are several LCE papers discussing about structural stability to the compression forces. (1) *Advanced Materials*, 2022, 34, 2200272. (2) *Advanced Healthcare Materials*, 2020, 9, 1901136. (3) *Advanced Materials*, 2020, 32, 2000797. (4) *ACS Appl. Mater. Interfaces* 2021, 13, 12698.

Answer: Your suggestion is greatly appreciated. The references in the manuscript have been revised as follows:

- The following references related to the mechanical performances of LCE structures are added:

Jeon, S., Shen, B., Traugutt, N. A., et al. Synergistic Energy Absorption Mechanisms of Architected Liquid Crystal Elastomers. *Adv. Mater.* **34** 2200272 (2022).

Volpe, R. H., Mistry, D., Patel, V. V., et al. Dynamically Crystallizing Liquid-Crystal Elastomers for an Expandable Endplate-Conforming Interbody Fusion Cage. *Adv. Healthcare Mater.* **9** 1901136 (2019).

- Ref 11 has been replaced with the following reference :

Kim, K., Guo, Y., Bae, J., et al. 4D Printing of Hygroscopic Liquid Crystal Elastomer Actuators. *Small* **17** 2100910 (2021).

- For the purpose of rigorous expression, we have removed the statement “rapid action” because the actuation rate of liquid crystals depends on external stimuli.

- Ref 12 has been replaced with the following reference:

Zeng, H., Wasylczyk, P., Parmeggiani, C., et al. Light-Fueled Microscopic Walkers. *Adv. Mater.* **27** 3883 (2015).

- The following reference is added:

Kim, K., Guo, Y., Bae, J., et al. 4D Printing of Hygroscopic Liquid Crystal Elastomer Actuators. *Small* **17** 2100910 (2021).

- Ref 17 and 18 has been replaced with the following reference:

Guin, T., Settle, M. J., Kowalski, B. A., et al. Layered liquid crystal elastomer actuators. *Nat. Commun.* **9** 2531 (2018).

- The purpose of Ref 35 is to point out that many 4D printed anisotropic materials have universality in deformation modes, not limited to LCE materials. Additionally, the following reference related to LCE is added:

He, Q., Wang, Z., Wang, Y., et al. Electrically controlled liquid crystal elastomer - based soft tubular actuator with multimodal actuation. *Sci. Adv.* **5** eaax5746 (2019).

- Ref 41 has been replaced with the following reference:

Yuan, C., Roach, D. J., Dunn, C. K., et al. 3D printed reversible shape changing soft actuators assisted by liquid crystal elastomers. *Soft Matter* **13** 5558 (2017).

- The author's name has been corrected.

- The following references are added:

Jeon, S., Shen, B., Traugutt, N. A., et al. Synergistic Energy Absorption Mechanisms of Architected Liquid Crystal Elastomers. *Adv. Mater.* **34** 2200272 (2022).

Volpe, R. H., Mistry, D., Patel, V. V., et al. Dynamically Crystallizing Liquid-Crystal Elastomers for an Expandable Endplate-Conforming Interbody Fusion Cage. *Adv. Healthcare Mater.* **9** 1901136 (2019).

Traugutt, N. A., Mistry, D., Luo, C., et al. Liquid-Crystal-Elastomer-Based Dissipative

Structures by Digital Light Processing 3D Printing. *Adv. Mater.* **32** 2000797 (2020).

Luo, C., Chung, C., Traugott, N. A., et al. 3D Printing of Liquid Crystal Elastomer Foams for Enhanced Energy Dissipation Under Mechanical Insult. *ACS Appl. Mater. Interfaces* **13** 12698 (2021).

Reviewer #4

Although the previous concerns have been well addressed, some minor questions should be answered. Here are specific points for consideration.

1. In Fig. 4f, the diameter of the composite filaments with various fiber off-center positions is not uniform, how to control the diameter?

Answer: This non-uniformity mainly occurs at the nodes of the truss structure. This is because the printing nozzle will stop in the suspended position after each filament is formed to cure the liquid crystal, and a small amount of liquid crystal will overflow from the nozzle. The subsequent solution is to design a device with stronger airtightness. This is our mistake in capturing and enlarging the non representative endpoint portion in the photo. Therefore, we adjusted the enlarged image in Figure 4f.

In most areas except for nodes, slight unevenness of 4D printing filament may occur due to the aggregation of liquid crystal droplets. The subsequent solution is to enhance the light curing efficiency or choose liquid crystal and fiber materials with better wetting effect.

2. The three structures in Fig. 6a, 6e, and 6i are axisymmetric, while the deformation in the simulation results (Fig. 6d, 6h, and 6l) is not axisymmetric. Are the structures deformed in three-dimensional space not two-dimensional plane?

Answer: I'm sorry that this figure caused you a misunderstanding. In fact, the simulation results of these structures are symmetrical. Due to the fact that the deformation in Figure 6i is three-dimensional, there is almost no significant change visible from a top view. In order to demonstrate the shape changes of the structure in three directions, these simulation results were not observed from a top view perspective, resulting in an appearance of seemingly asymmetric deformation.

REVIEWERS' COMMENTS

Reviewer #3 (Remarks to the Author):

The authors made a considerable effort to improve the manuscript.
The concerns from the reviewer have been resolved.

Reviewer #4 (Remarks to the Author):

The authors have addressed all comments. I recommend accepting the paper in its current form.